# An optogenetic toolkit for light-inducible antibiotic resistance

Michael B. Sheets[1,2], Nathan Tague[1,2] & Mary J. Dunlop [1,2] ✉

Antibiotics are a key control mechanism for synthetic biology and microbiology. Resistance genes are used to select desired cells and regulate bacterial populations, however their use to-date has been largely static. Precise spatiotemporal control of antibiotic resistance could enable a wide variety of applications that require dynamic control of susceptibility and survival. Here, we use light-inducible Cre recombinase to activate expression of drug resistance genes in *Escherichia coli*. We demonstrate light-activated resistance to four antibiotics: carbenicillin, kanamycin, chloramphenicol, and tetracycline. Cells exposed to blue light survive in the presence of lethal antibiotic concentrations, while those kept in the dark do not. To optimize resistance induction, we vary promoter, ribosome binding site, and enzyme variant strength using chromosome and plasmid-based constructs. We then link inducible resistance to expression of a heterologous fatty acid enzyme to increase production of octanoic acid. These optogenetic resistance tools pave the way for spatiotemporal control of cell survival.

Antibiotic resistance genes are widely used in synthetic biology. They are included in genetic constructs to ensure plasmid propagation. Resistance genes also play an important role in cloning methods. Examples include chromosomal insertions, where expression of resistance genes can be used as a selective marker for successful integration[1], or in the creation of transposon libraries, where drug resistance is used as an intermediate selection mechanism before being swapped for an alternative sequence[2,3].

Although antibiotic resistance genes are a staple of synthetic biology and microbial biotechnology research, there are few methods for dynamic control of their expression. The ability to control drug resistance spatially and temporally could open new avenues for synthetic biology research. As an analogy, when Sheth et al.[4] developed an inducible origin of replication—another ubiquitous feature within synthetic biology constructs—it sparked new areas of research including biological data storage[5] and whole-cell riboswitch diagnostics[6]. Spatiotemporal control over drug resistance could enable spatial patterning in living biomaterials[7], selection of single cells from microfluidic systems[8,9], and improved understanding of the role dynamics play in clinical antibiotic resistance[10]. For example, resistance is often spread through horizontal gene transfer events[11,12],

which are difficult to monitor and control at the single-cell level. New systems for control offer the potential for future studies quantifying how different spatiotemporal arrangements of cells acquiring resistance can lead to population-level proliferation or collapse.

Optogenetic methods are a powerful and widely used tool for controlling gene expression[13]. The delivery of light to cells can be regulated in space and time, and can be integrated directly into computational workflows[14,15]. Optogenetic systems in bacteria have been used to control gene expression for a variety of applications[13,16], including to drive metabolic flux[17], regulate the gut microbiome[18], control cell morphology[19], and regulate co-culture dynamics[20]. Using light to control cell survival has been a focus of microbial engineering across species. For example, optogenetic regulation has been used to control nourseothricin resistance in *Saccharomyces cerevisiae*[21] and bleomycin resistance in *Yarrowia lipolytica*[22]. In *Escherichia coli*, light has been used to control antibiotic resistance via individually designed photo-caged antibiotics[23] or by leveraging the natural photosensitivity of tetracycline[24]. However, because these methods require careful protein engineering or exploit properties specific to a single drug, they do not easily generalize across different resistance mechanisms. An alternative approach used a light-inducible promoter to reversibly

[1]Department of Biomedical Engineering, Boston University, Boston, MA 02215, USA. [2]Biological Design Center, Boston University, Boston, MA 02215, USA. ✉e-mail: mjdunlop@bu.edu

control chloramphenicol resistance[20,25]. Such a method could generalize to other resistance genes, however experiments were limited to control of the chloramphenicol acetyltransferase enzyme. The ideal platform for light-inducible resistance would be both generalizable for different antibiotic resistance genes and tunable across antibiotic concentrations to flexibly enable diverse studies in synthetic biology and microbiology.

To address these needs, we used the blue light-inducible Cre recombinase OptoCreVvd2 to activate antibiotic resistance genes[26]. Using this system, we excise a *loxP*-flanked transcription terminator

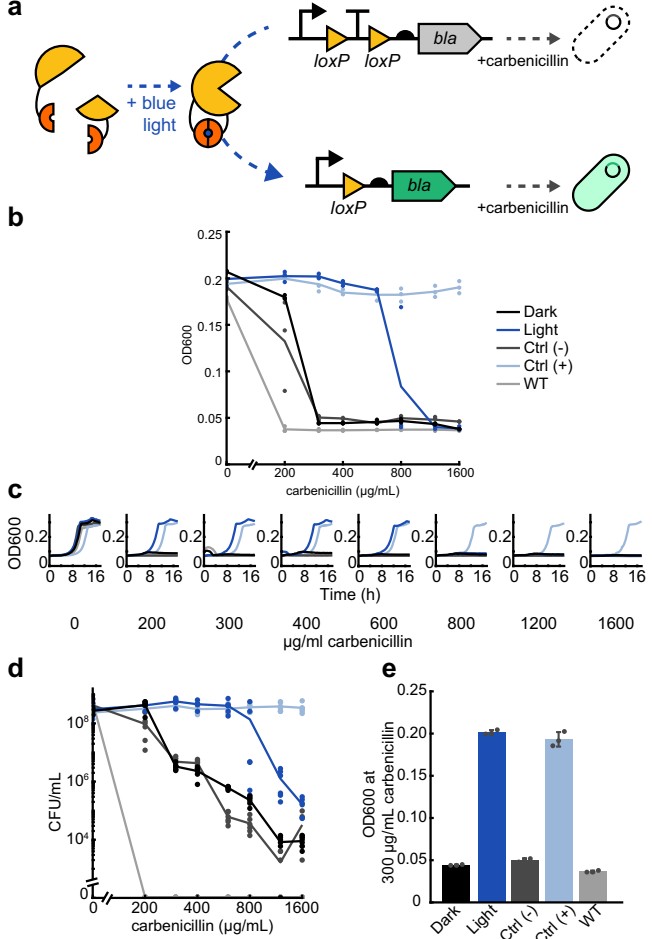

**Fig. 1 | Optogenetic activation of beta-lactamase antibiotic resistance using OptoCre-*bla*. a** Split Cre recombinase fragments are linked to blue light-inducible Vvd photodimer domains. When exposed to blue light, Cre becomes active and can excise a transcription terminator between two *loxP* domains, allowing increased expression of beta-lactamase (*bla*). The expression of OptoCre-*bla* then allows cells to survive in the presence of the antibiotic carbenicillin. **b** Minimum inhibitory concentration (MIC) curves of chromosomally-integrated OptoCre-*bla* constructs grown in carbenicillin for 18 h. Light-induced samples were exposed to blue light for 2 h immediately before exposure to carbenicillin. Growth measured by OD600 (*n* = 3). The strains in both the dark and light conditions contain the resistance induction construct and OptoCreVvd2. Control (−) cells contain the resistance activation construct but no Cre recombinase. Control (+) cells contain a constitutively expressed *bla* gene. Wild-type (WT) cells are MG1655 without modification or plasmids. **c** Time-course growth of OptoCre-*bla* resistance activation constructs across different concentrations of carbenicillin. **d** Colony forming unit (CFU) counts of cultures from MIC data (*n* = 6). After growth in carbenicillin for 18 h, samples were spotted on agar plates and colonies were counted the next day. **e** Optimal OptoCre-*bla* activation conditions. Resistance activation constructs grow in 300 μg/mL carbenicillin after exposure to blue light for 2 h, but not when kept in the dark. Growth is quantified by OD600 after 18 h. Error bars show standard deviation around the mean (*n* = 3 biological replicates).

between a gene and promoter, allowing for increased gene expression after exposure to 465 nm blue light (Fig. 1a). Recombinase technology has been used successfully for a variety of applications that require robust and inducible control of gene expression, including gene logic circuits and cell lineage tracking[27–29]. We selected this system for its relatively short activation time, flexibility in construct design (requiring only the addition of *loxP* sites), and minimal basal expression in uninduced cells[26]. In addition, the permanent OFF-ON switch caused by Cre allows for selection of resistant cells at any point after light exposure, allowing for cellular memory after the light input has been removed. Although this irreversibility does not allow for complex temporal dynamics, one-time induction provides benefits that may be advantageous in certain applications such as allowing irreversible activation before a culture becomes too dense for light penetration, or minimizing exposure in light-sensitive strains.

Here, we used Cre to induce expression of four antibiotic resistance genes, which we selected for their ubiquity in synthetic biology applications as well as their range of mechanisms of action (Table 1). Specifically, we chose the carbenicillin/ampicillin resistance gene beta-lactamase (*bla*), which is both clinically relevant and widely used in synthetic biology. Beta-lactam antibiotics inhibit cell wall biosynthesis, and are enzymatically degraded by the beta-lactamase enzyme[12,30,31]. We also included kanamycin nucleotidyltransferase (*knt*), which provides enzymatic resistance against kanamycin, an antibiotic that causes mistranslation by the 30S ribosomal subunit[32]. Chloramphenicol acetyltransferase (*cat*) provides enzymatic resistance against chloramphenicol, which interferes with the 50S ribosomal subunit to cause protein synthesis to stall[33]. Lastly, we included the tetracycline efflux pump A (*tetA*) as a non-enzymatic, efflux-based resistance mechanism[34]. Tetracycline binds the 30S ribosomal subunit to inhibit protein synthesis[35]. These four antibiotics include both bactericidal (carbenicillin/ampicillin and kanamycin) and bacteriostatic (chloramphenicol and tetracycline) drugs. This selection of resistance mechanisms shows both the broad variety of mechanisms that can be controlled using this system and introduces multiple options for synthetic tools that can be incorporated into existing bacterial systems.

When controlling expression of antibiotic resistance genes, key performance metrics include the uninduced and induced expression levels. For example, when using potent enzymes like beta-lactamases, even a small amount of basal expression can allow bacterial growth in the presence of an antibiotic. Moreover, expression in the induced state should also be sufficient to provide resistance at drug concentrations comparable to typical working ranges for the antibiotic, which could further vary between use cases. To optimize these two features in our platform, we varied the gene copy number, promoter, ribosome binding site (RBS), and coding sequence to tune the minimum inhibitory concentration (MIC) of antibiotic at which cells survive after exposure to blue light, while maintaining basal expression levels that are low enough to avoid erroneously triggering survival. We further demonstrated live activation of resistance genes and characterized cellular responses using single-cell time-lapse microscopy.

Finally, we demonstrated the utility of light-inducible resistance in a biotechnology application by co-expressing resistance genes with the heterologous thioesterase *Cp*FatB1 to increase production of octanoic acid, a medium-chain fatty acid. Medium-chain fatty acids are high-value biochemicals used in fuels, polymer production, flavorings, and fragrances, making them key metabolic engineering targets[36,37]. However, induction of *Cp*FatB1 is metabolically taxing, a common issue with expression of heterologous enzymes for bioproduction applications. This challenge has prompted researchers to develop systems where pathway expression timing can be precisely tuned to balance the tradeoff between growth and production[38,39]. For example, light induction has been used to increase production of other chemicals in *E. coli*, such as mevalonate and isobutanol[40]. In our system with OptoCreVvd2, light induction of *Cp*FatB1 coupled with antibiotic

**Table 1 | Summary of antibiotics and corresponding resistance genes used in this work**

| Antibiotic | Effect | Target | Resistance | Mechanism |
|---|---|---|---|---|
| Carbenicillin/Ampicillin | Bactericidal | Cell wall | bla | Enzymatic |
| Kanamycin | Bactericidal | Ribosome (30S) | knt | Enzymatic |
| Chloramphenicol | Bacteriostatic | Ribosome (50S) | cat | Enzymatic |
| Tetracycline | Bacteriostatic | Ribosome (30S) | tetA | Efflux pump |

selection for high expression of the heterologous enzyme significantly increased fatty acid production over light induction of *Cp*FatB1 alone.

This toolkit of light inducible resistance genes supports and extends the long use of antibiotics as cellular control mechanisms in synthetic biology, adding a spatial and temporal control mechanism to existing systems and setting the stage for future applications where light is used in combination with antibiotics to enable flexible control of cell behavior and survival.

## Results

### Light induction of beta-lactamase resistance
For optogenetic control of resistance genes, we used the blue light-inducible split Cre recombinase OptoCreVvd2[26]. This system allows for excision of genetic elements placed between *loxP* sites when cells are exposed to blue light. Excision can be completed in ~2 h, which is comparable to or faster than many existing bacterial optogenetic systems[19,25,41,42]. We used OptoCreVvd2 to excise a transcription terminator placed inside *loxP* sites between a promoter and an antibiotic resistance gene, allowing for expression of the resistance gene only after exposure to blue light.

We first used this system to control transcription of the beta-lactamase (*bla*) resistance gene (which we denote 'OptoCre-*bla*', Fig. 1a). Beta-lactam antibiotics, including ampicillin and carbenicillin, inhibit peptidoglycan layer biosynthesis in the bacterial cell wall. Beta-lactamase enzymes can inactivate beta-lactam antibiotics by hydrolyzing the beta-lactam ring on the antibiotic[43]. For these studies, we used the TEM-116 beta-lactamase, which is commonly used in antibiotic resistance cassettes for plasmid selection[44]. We integrated this genetic construct after *nupG* in the *E. coli* MG1655 chromosome.

To measure light-induced antibiotic resistance, we exposed cultures of OptoCre-*bla* to blue light for 2 h, then grew them overnight in the presence of carbenicillin and compared growth to cultures kept in the dark. We observed blue light-dependent differences in cell proliferation, where the MIC necessary to prevent growth was 300 µg/mL for cultures kept in the dark and 1200 µg/mL for cultures exposed to blue light (Fig. 1b). Negative control (− Control) cells with only the reporter and no Cre recombinase had comparable survival to cells with the full construct grown in the dark, indicating low expression of *bla* in the uninduced state. Positive control (+ Control) cells with constitutive expression of *bla* from its native promoter and RBS grew in all concentrations of carbenicillin tested, including at levels above the light-inducible strain, as expected for a fully resistant strain expressing the resistance gene in its native context. We further included *E. coli* MG1655 as a wild-type negative control, which did not grow in any concentration of carbenicillin used here.

To confirm that blue light-induced cells grow at rates comparable to the positive control strain, we collected time-series data demonstrating normal growth rates under a broad range of carbenicillin concentrations for cells grown in blue light, while cells without light induction failed to grow (Fig. 1c). We further validated the optical density-based MIC data by using colony forming unit (CFU) counts after antibiotic exposure (Fig. 1d). Although MIC data is an accurate assessment of cell growth in the presence of antibiotic, beta-lactam antibiotics also cause cell filamentation, which can increase optical density (OD) readings even when cells are not dividing, making *bla* resistance specifically important to confirm by CFU measurement[45].

However, our results using CFU counts confirm that the optical density measurements also translate to a clear difference in cell survival[46]. Overall, we found that OptoCreVvd2 can be used to induce beta-lactamase resistance and we identified concentrations of carbenicillin with robust differences between dark and blue-light activated expression of the *bla* resistance gene construct (Fig. 1e and Table 2).

### Optimization of kanamycin resistance
We next sought to generalize this system to other antibiotic resistance genes. While the excision of a terminator can easily couple light to the expression of an antibiotic resistance gene, this is not the same as light-inducible survival. Control of survival requires no or low expression of the antibiotic resistance gene in the dark, such that cells remain susceptible to antibiotics. The design also requires that induction of the resistance gene is sufficient to confer resistance. The thresholds for these two features can vary dramatically with different antibiotic resistance mechanisms, which have different rates of antibiotic degradation and export. Thus, while previous work in our lab has shown that OptoCreVvd2 can control expression of a fluorescent protein with low basal expression and a 10-fold change with light[26], naively replacing the fluorescence gene with an antibiotic resistance gene may not produce the desired behavior. Therefore, we defined a general process for adapting the OptoCreVvd2 system to different antibiotic resistance genes (*bla, knt, cat,* and *tetA*) and were able to show survival at customized ranges of antibiotic concentration.

To generalize our system to other antibiotics, we began by exchanging *bla* for *kanamycin nucleotidyltransferase* (*knt*) in our induction construct to make OptoCre-*knt* (Fig. 2a). The antibiotic kanamycin causes mistranslation by binding to the 30S subunit of the bacterial ribosome. The *knt* enzyme catalyzes transfer of a nucleotide to kanamycin, inactivating the antibiotic[32]. Although our initial design of OptoCre-*knt* did show resistance activation using light, the kanamycin concentration required to see this difference was very high – over 1000 µg/mL (Fig. 2b), compared to 25–50 µg/mL commonly used for plasmid propagation[24,44]. Although matching common working concentrations of antibiotics is not essential, using concentrations in the vicinity of these ranges provides benefits including the ability to study community effects at physiological concentrations and limiting overall antibiotic needs.

Thus, we set out to tune gene expression through optimization of the genetic architecture surrounding the gene. To lower basal expression, we weakened the promoter or RBS driving *knt* expression. By changing the promoter and RBS of OptoCre-*knt*, we were able to shift the expression of the resistance gene to allow survival at antibiotic concentrations much closer to the MIC of wild-type MG1655 (Fig. 2b). We tested a range of promoter and RBS combinations to show how these alterations impact survival at varying antibiotic concentrations. We used constitutive promoters of varying strength all based on the T7A1 viral promoter, ranging from medium, P, to medium-low, P*, to low, P**, transcriptional strength. We also used the RBS of gene 10 in the T7 phage[47], which we denote R, and a RBS that we computationally designed to be weaker[48], which we denote R* (Fig. 2a). Changing P to P* decreased the MIC for the dark state to 200 µg/mL kanamycin, while P** further reduced it to 150 µg/mL (Fig. 2b). The MIC for the light state was also reduced, as expected, but still maintained a wide range of kanamycin concentrations resulting in survival. With P*,

**Table 2 | Resistance activation constructs used in this study and respective dark and light state MICs**

| Resistance | Site | Promoter | RBS | Dark MIC (μg/mL) | Light MIC (μg/mL) |
|---|---|---|---|---|---|
| **OptoCre-*bla*** | | | | | |
| | **chromosomal (*nupG*)** | **P** | **R** | **300** | **1200** |
| | plasmid (p15A origin) | P | R | 10000 | 10000 |
| **OptoCre-*knt*** | | | | | |
| | chromosomal (*nupG*) | P | R | 1600 | 2400 |
| | **chromosomal (*nupG*)** | **P\*** | **R** | **200** | **1200** |
| | chromosomal (*nupG*) | P\*\* | R | 200 | 600 |
| | chromosomal (*nupG*) | P | R\* | 4 | 6 |
| | plasmid (p15A origin) | P | R | 6400 | 9600 |
| | plasmid (p15A origin) | P\* | R | 4800 | 6400 |
| | plasmid (p15A origin) | P\*\* | R | 4800 | 6400 |
| | plasmid (p15A origin) | P | R\* | 15 | 40 |
| **OptoCre-*cat*** | | | | | |
| | chromosomal (*nupG*) *cat* | P | R | 15 | 30 |
| | chromosomal (*nupG*) *cat*$_{T172A}$ | P | R | 10 | 30 |
| | plasmid (p15A origin) *cat* | P | R | 150 | 300 |
| | **plasmid (p15A origin) *cat*$_{T172A}$** | **P** | **R** | **75** | **200** |
| **OptoCre-*tetA*** | | | | | |
| | chromosomal (*nupG*) | P | R | 1.5 | 1.5 |
| | chromosomal (*nupG*) | P$_{tet}$ | R$_{tet}$ | 1 | 2 |
| | plasmid (p15A origin) | P | R | 1.5 | 1.5 |
| | **plasmid (p15A origin)** | **P$_{tet}$** | **R$_{tet}$** | **4** | **8** |

Optimal constructs for each antibiotic are indicated in bold, where this selection considers both fold change and total numerical difference in MIC on light exposure.

kanamycin levels between 200 and 800 μg/mL resulted in light-induced survival, while for P\*\* the range was from 150 to 500 μg/mL. Using R\* in combination with P caused a dramatic decrease in both the dark state MIC and the effective concentrations for light-induced survival, resulting in a narrow range between 4 and 6 μg/mL kanamycin.

Although the chromosomally integrated constructs used so far have the advantages of low background expression and do not require a selection marker, plasmids offer their own advantages for light-inducible resistance systems. Many resistance genes are naturally found on plasmids, and a plasmid origin allows for convenient transfer of systems between different strains. We further characterized our constructs on plasmids containing the p15A origin of replication, which has approximately ten copies per cell (Fig. 2c)[49]. Changing from chromosomal integration to a p15A plasmid increased the range of antibiotic concentrations at which cells containing OptoCre-*knt* selectively survive by over 5-fold. Despite this increase, we found that strategies such as lowering promoter or RBS strength can have a counterbalancing effect. We also characterized a p15A plasmid-based OptoCre-*bla*, however its basal resistance was too high to be considered functional (Supplementary Fig. 1a). Overall, the plasmid-based system with OptoCre-*knt* removes the need for the chromosomal insertion process, increasing the ease at which these constructs can be used in different strains or contexts.

The flexibility afforded by these different designs led us to develop multiple constructs, and the optimal construct is likely to be application-specific. For example, the lower concentrations shown here are near the wild-type MIC, which is optimal for studies looking to characterize resistance acquisition using phenotypically-relevant antibiotic concentrations. In contrast, the higher concentrations allow more stringent cell selection for studies where only the activated cells should survive. Through this optimization process, we found constructs that allow a greater kanamycin MIC fold change between dark and light-exposed cultures compared to our original construct,

notably P\* and R driving OptoCre-*knt* expression on the chromosome is an ideal example of our optimized design (Fig. 2d and Table 2).

**Protein-level optimization of chloramphenicol resistance**

Light-induced survival requires a tight OFF-state where cells are susceptible to antibiotic. As we have demonstrated, this can be achieved with low basal expression of the resistance gene. However, the uninduced state can also be minimized if the resistance enzyme itself is weaker. Thus, when activating the chloramphenicol acetyltransferase (*cat*) enzyme, we took advantage of a known mutation to decrease the strength of the enzyme itself. Chloramphenicol prevents protein synthesis by binding to the 50S ribosomal subunit, where it inhibits peptide bond formation. The *cat* enzyme prevents chloramphenicol from binding to the ribosome by attaching an acetyl group from acetyl-CoA to the antibiotic[50]. By using the weaker *cat*$_{T172A}$ variant[51,52], we lowered the concentration of antibiotic at which cells survive (Fig. 3a). In the OptoCre-*cat* design, we used the promoter P and RBS R, and compared light-induced survival by *cat* and *cat*$_{T172A}$. Here we observed a sharper decrease in dark OFF-state resistance with *cat*$_{T172A}$ compared to *cat*, lowering basal resistance to that of the wild-type strain on a chromosomally integrated construct (Fig. 3b). We also observed a decrease in MIC values for *cat* and *cat*$_{T172A}$ on a p15A plasmid origin (Fig. 3c). This enzyme mutant approach to optimization may be particularly helpful when working with enzymes that show resistance to high concentrations of antibiotic even with minimal basal gene expression, or if using this system on a plasmid with a high copy number where it is hard to limit basal expression. This approach creates another point at which resistance levels can be fine-tuned, and the light-induced growth difference shown by *cat*$_{T172A}$ on a p15A plasmid origin is a particularly versatile optimized design (Fig. 3d and Table 2).

We wondered whether it would be possible to tune resistance levels by adjusting light exposure properties. To test this, we further characterized the OptoCre-*cat* design by modifying light exposure duration and intensity (Supplementary Fig. 2). We found that

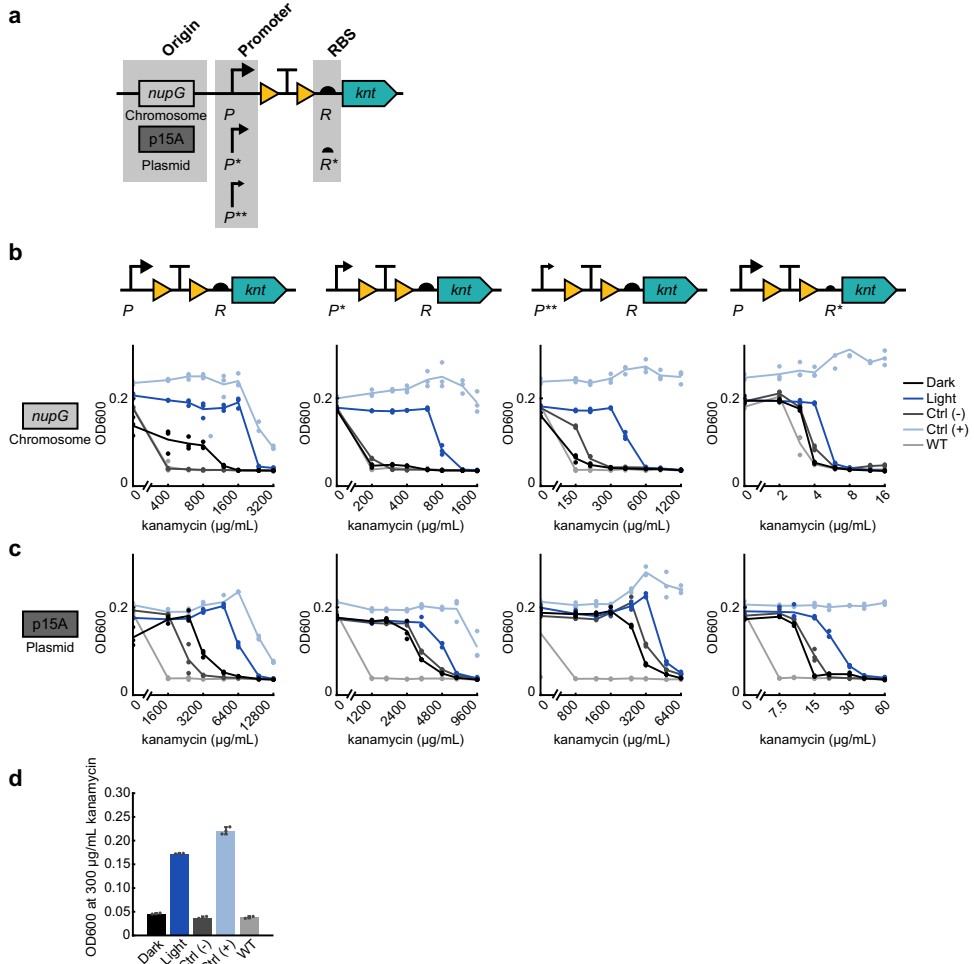

**Fig. 2 | Optimizing optogenetic activation of kanamycin resistance from OptoCre-*knt* by modifying promoter and ribosome binding site (RBS).** **a** Expression levels of the kanamycin resistance gene *knt* can be tuned by changing the strength of the promoter and RBS, as well as the origin of replication. Promoter strength ranges from P (medium) to P* (medium-low) to P** (low). RBS strength ranges from R (strong) to R* (weak). **b** MIC curves of OptoCre-*knt* activation cassettes on the chromosome, with promoter P, P*, or P** and RBS R or R* ($n = 3$). **c** MIC curves of OptoCre-*knt* activation cassettes on a plasmid with the p15A origin of replication ($n = 3$). **d** Optimal OptoCre-*knt* activation conditions, using P* and R on the chromosome at 300 μg/mL kanamycin. Growth is quantified by OD600 after 18 h. Error bars show standard deviation around the mean ($n = 3$ biological replicates).

resistance levels were tunable, and depended on a combination of both duration of light exposure and the intensity of the blue light.

A common issue with many of our designs is that basal expression results in resistance levels that, although low, still exceed those observed in the wild-type strain. In principle, this leakiness could be the result of several different issues including spontaneous recombination events, mutations in the terminator, or readthrough of the terminator. We sequenced the *loxP* and terminator region of the (-) Control strain without Cre from both 0 μg/mL chloramphenicol and 75 μg/mL chloramphenicol conditions, where the latter represents the highest antibiotic concentration condition that showed any growth. Sequencing results from both conditions matched the original sequence of the plasmid, confirming that leaky expression is not the result of spontaneous recombination or terminator mutations. These results are in line with characterizations of the original OptoCreVvd2 design, which showed consistent low basal expression of a fluorophore with no evidence of spontaneous recombination[26]. To mitigate the potential for terminator readthrough we then attempted to lower basal expression by swapping the strong BBa_B0015 terminator from our original design to the synthetic terminator L3S2P21, which was the strongest terminator identified in an extensive set characterized in Chen et al.[53]. However, this did not show further improvement over the terminator used in our initial design (Supplementary Fig. 3), so we did

not pursue this avenue further. For applications where minimal basal expression is critical, alternative designs could test other terminators[53,54] or different circuit design approaches[55–57].

## Efflux pump enabled tetracycline resistance

Light induction can also be applied to non-enzymatic antibiotic resistance mechanisms, such as the *tetA* efflux pump. The antibiotic tetracycline reversibly binds to the 30S ribosomal subunit, inhibiting protein synthesis. The *tetA* efflux pump localizes to the inner membrane and exports magnesium-tetracycline chelate complexes by importing a proton[35]. In sharp contrast to *bla*, *knt*, and *cat* where the native resistance levels were high and our engineering efforts aimed at reducing potency, we found that our initial design for inducible *tetA* did not show resistance over wild-type MG1655 when expressed with promoter P and RBS R on a p15A origin plasmid (Supplementary Fig. 1b), conditions which produced the highest levels of resistance in the constructs we tested previously. To compensate for this, we opted to use the strong native promoter $P_{tet}$ with its corresponding RBS $R_{tet}$ to allow full expression of the *tetA* gene[58] to create OptoCre-*tetA* (Fig. 4a). Using this native architecture, OptoCre-*tetA* showed some activation when chromosomally integrated (Fig. 4b), and exhibited strong activation on the p15A plasmid origin (Fig. 4c). Notably, the dark-state basal resistance over wild-type MG1655 was minimal,

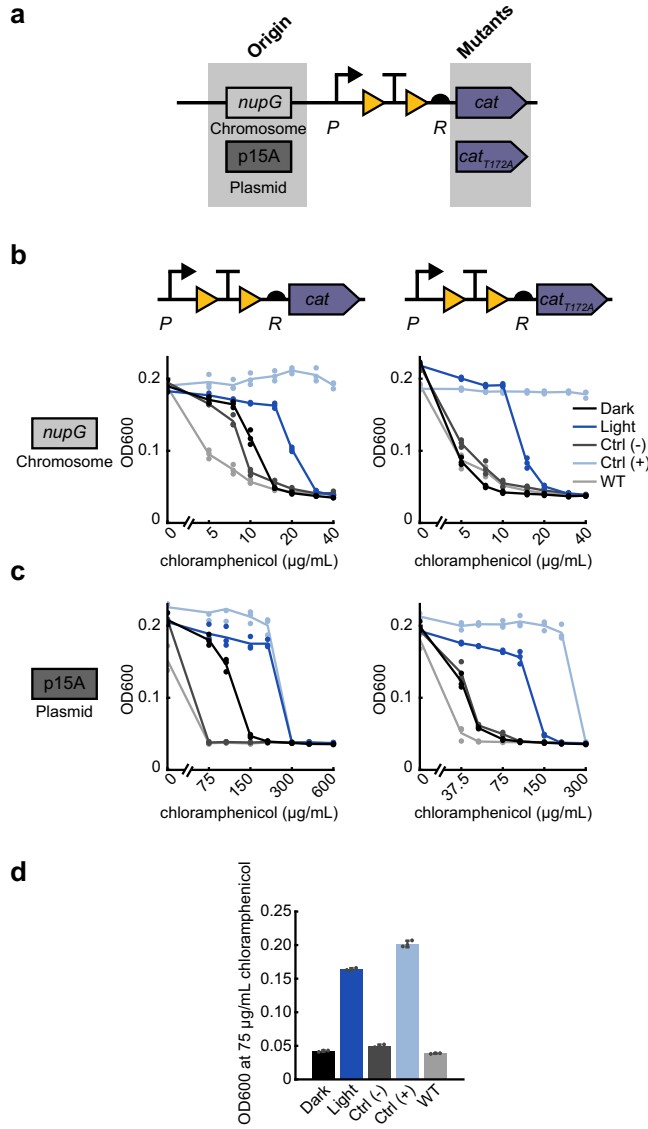

**Fig. 3 | Tuning optogenetic activation of chloramphenicol resistance from OptoCre-*cat* through decreased enzymatic activity. a** Resistance given by the chloramphenicol resistance gene *cat* can be lowered by using the *cat*$_{T172A}$ variant on a chromosomal or plasmid origin. **b** MIC curves of OptoCre-*cat* activation cassette with the native enzyme *cat* or weakened enzyme *cat*$_{T172A}$ on the chromosome (*n* = 3). **c** MIC curves of the *cat* and *cat*$_{T172A}$ activation cassettes on a plasmid with the p15A origin (*n* = 3). **d** Optimal activation conditions, using *cat*$_{T172A}$ with promoter P and RBS R on a plasmid origin at 75 μg/mL chloramphenicol. Growth is quantified by OD600 after 18 h. Error bars show standard deviation around the mean (*n* = 3 biological replicates).

allowing for activation of OptoCre-*tetA* at low tetracycline concentrations. Thus, we found that the p15A plasmid-based version is an ideal construct for tetracycline resistance (Fig. 4d and Table 2).

## Single-cell microscopy showing resistance activation

The spatial and temporal precision enabled by optogenetics allows these constructs to be used for a variety of applications, including single-cell studies of bacterial antibiotic resistance. How resistance acquisition leads to bacterial survival at the single-cell level is of particular interest in the context of horizontal gene transfer. Existing single-cell horizontal gene transfer studies have previously characterized gene transfer rates and shown important connections to quorum sensing[59,60]. However, natural instances of horizontal gene transfer are

infrequent and difficult to control in space and time, especially relative to antibiotic exposure. Previous studies have also shown that stochastic acquisition of resistance in single cells is not always enough to cause the proliferation of phenotypically resistant cells[61]. To study when horizontal gene transfer events lead to the spread of resistance in populations, it would be interesting to model a single cell's acquisition of resistance with optogenetic control of antibiotic susceptibility. This could be used to characterize when and how resistance acquisition in single cells leads to antibiotic evasion, and how specific antibiotic dosing schedules and concentrations impact evasion frequency.

Here we show a proof-of-concept for the first steps in this class of studies by inducing cell growth using blue light for cells on agarose pads. Using time-lapse microscopy, we placed cells containing light-activatable antibiotic resistance on agarose pads containing antibiotics and compared the growth of cells that were kept in the dark to those exposed to blue light. For these studies we elected to focus on a subset of resistance genes, selecting kanamycin and chloramphenicol as examples of bactericidal and bacteriostatic antibiotics, respectively. We characterized resistance activation for chromosomally integrated OptoCre-*knt* resistance to kanamycin (Fig. 5a and Supplementary Movie 1), and plasmid-based OptoCre-*cat* resistance to chloramphenicol (Fig. 5b and Supplementary Movie 2). We found that cells with light-induced resistance showed a short lag before growth compared to their constitutively resistant positive controls, which likely corresponds to the time needed to excise the transcription terminator and allow expression of the resistance gene. In contrast, when cells were kept in the dark, growth was inhibited and we observed examples of loss of membrane integrity (Supplementary Movie 1–2). To quantify recovery of resistance-induced cells, we calculated the percentage of cells in the initial frame that recovered over the course of the movie (Supplementary Fig. 4). We defined the time of recovery as the point at which a cell first divides, and found 70% recovery for OptoCre-*knt* and 42% recovery for OptoCre-*cat* with light exposure (compared to 13% and 4% for cells in the dark, though notably cells in the dark rarely experienced more than one division event, Supplementary Movie 1–2). Although these data show clear differences between light and dark exposure, the incomplete rates of recovery under light exposure may be due to simultaneous exposure to antibiotic and light, likely rendering some cells nonviable before they can be induced. In the future, microfluidic experiments could help to assess recovery rates as a function of relative light induction and antibiotic addition timing.

Looking towards future applications that require the control of subpopulations of cells, we also used a digital micromirror device (DMD)[62] to activate resistance in a subset of cells. The DMD allows for programmed illumination of specific areas within a field of view. We illuminated half of the field of view and saw preferential activation of cells in the blue light illuminated region (Supplementary Fig. 5). We did observe some growth in a subset of the cells in the dark half of the field of view, albeit at reduced levels relative to the illuminated half. This may be due to diffusion of light from the DMD, as we do not see this activation until after DMD illumination starts. Compared to the longer, lower intensity exposures we used in liquid culture experiments, the DMD is designed to deliver intense periods of light. These differences suggest that DMD activation protocols and setups could be optimized for improved activation timing in the future.

## Improving octanoic acid production using antibiotic resistance selection

To demonstrate the utility of the system for biotechnology applications, we next focused on increasing yield of octanoic acid through co-expression of *cat* or *tetA* with the thioesterase *Cp*FatB1 (Fig. 6a-b). *Cp*FatB1 is derived from the plant species *Cuphea palustris* and has been optimized for expression in *E. coli*[63]. *Cp*FatB1 expression is taxing on cells, therefore directly coupling its induced expression with

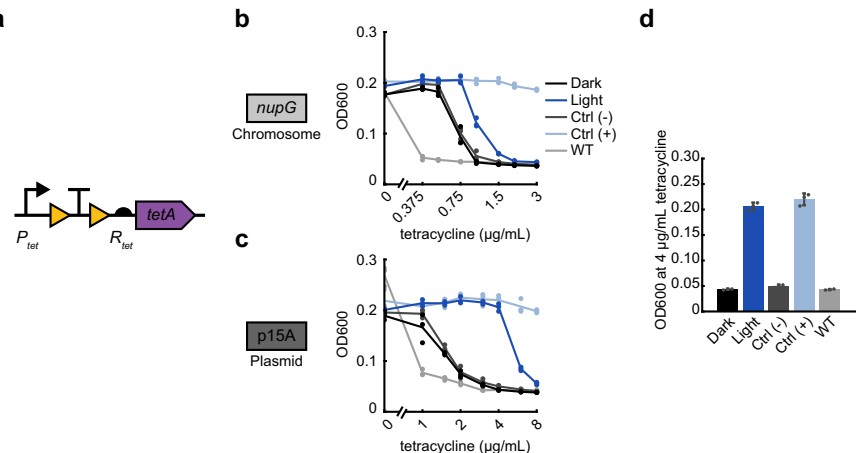

**Fig. 4 | Optogenetic activation of efflux-based tetracycline resistance with OptoCre-*tetA*. a** Tetracycline resistance gene *tetA* is expressed using the native promoter P$_{tet}$ and native RBS R$_{tet}$, to allow maximal expression of the gene. **b** MIC curves of OptoCre-*tetA* on the chromosome and **c** p15A plasmid origin (*n* = 3).

**d** Optimal OptoCre-*tetA* activation conditions, using the p15A plasmid origin at 4 μg/mL tetracycline. Growth is quantified by OD600 after 18 h. Error bars show standard deviation around the mean (*n* = 3 biological replicates).

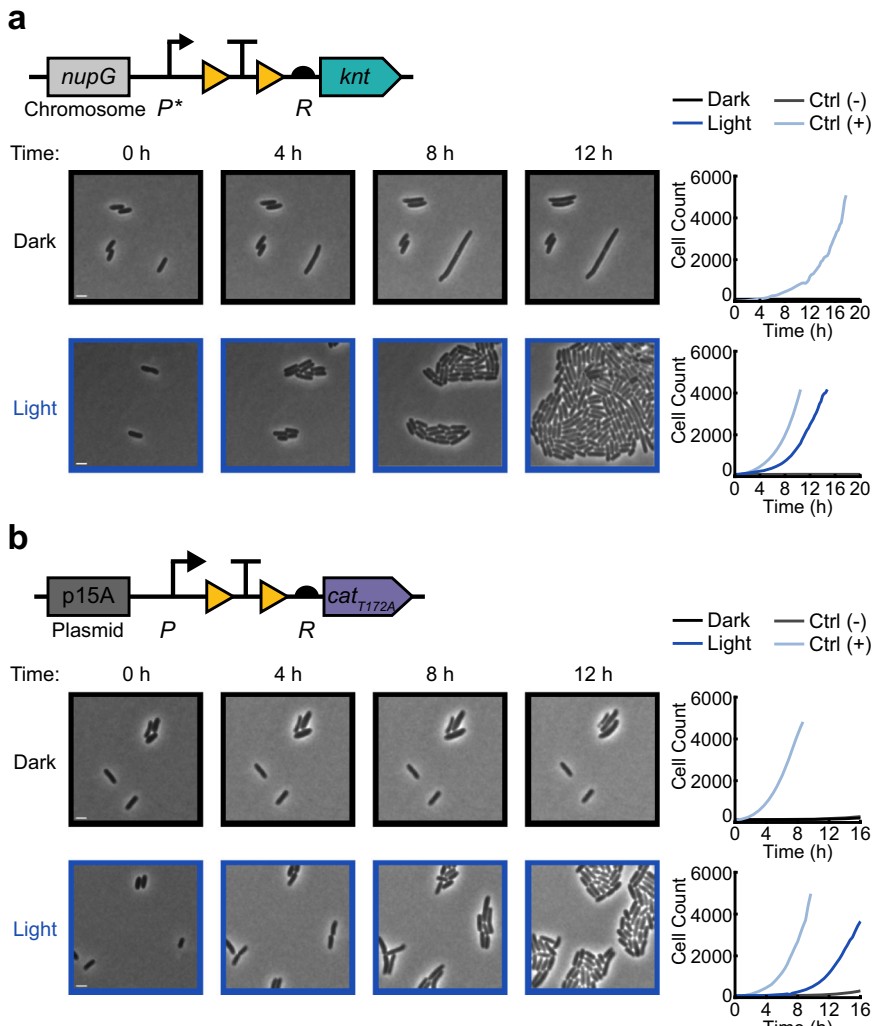

**Fig. 5 | Single-cell time-lapse microscopy of light-induced antibiotic resistance.** Activation of **a** the chromosomal OptoCre-*knt* resistance construct with promoter P* and RBS R on agarose pads containing 400 μg/mL kanamycin, and **b** the p15A plasmid-based OptoCre-*cat* resistance construct using *cat*$_{T172A}$ with promoter P and RBS R on agarose pads containing 60 μg/mL chloramphenicol. Microscopy images show representative samples of the resistance activation strains in the dark or with blue light (scale bar = 2 μm). Cell counts over time are cumulative across multiple imaging positions for each condition, with each plot containing the OptoCre resistance strain along with negative and positive controls (*n* = 3 biological replicates).

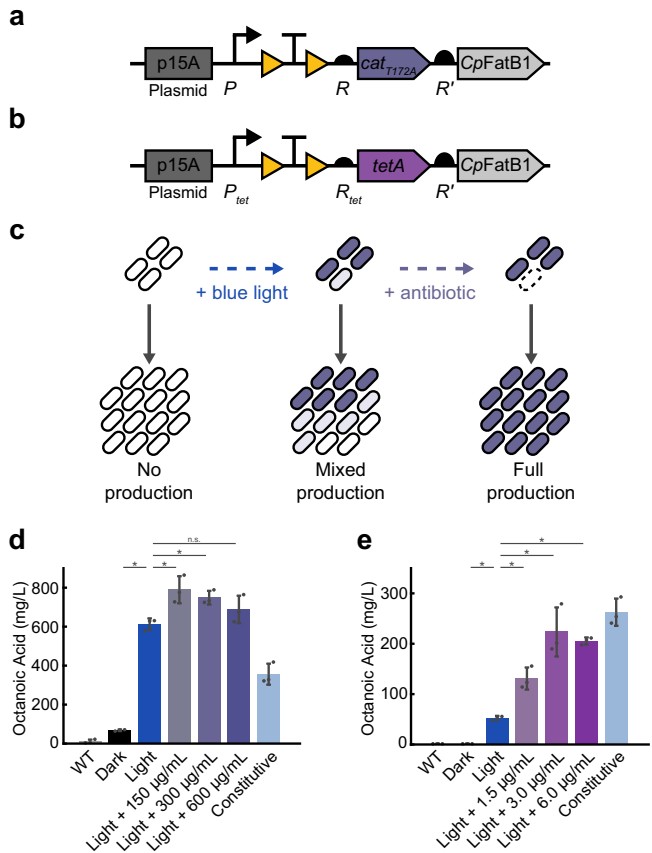

**Fig. 6 | Light-inducible production of octanoic acid. a** Activation of OptoCre-*cat* using *cat*$_{T172A}$ or **b** OptoCre-*tetA* is coupled with *Cp*FatB1 by introducing the gene downstream of the drug resistance marker under the strong RBS R'. **c** Blue light induces expression of the resistance gene and *Cp*FatB1, but does not guarantee all individuals within a community are expressing the genes. Non-producers have a growth advantage due to the burden of *Cp*FatB1. Addition of chloramphenicol prevents growth of any individuals not producing the resistance gene and consequently *Cp*FatB1. **d** Octanoic acid production coupled with OptoCre-*cat* measured by GC-MS. **e** Octanoic acid production coupled with OptoCre-*tetA* measured by GC-MS. Significance was determined using a two-tailed Welch's t-test: *$P < 0.05$; n.s. not significant. Error bars show standard deviation around the mean ($n = 3$ biological replicates).

could further improve yields. We found that with OptoCre-*cat*, adding 150 μg/mL chloramphenicol significantly improved production, increasing it by 29% over the light induction-only condition, with higher chloramphenicol concentrations resulting in similar performance (Fig. 6d). Light induction alone or with antibiotic also showed a substantial improvement in yield over a constitutively expressed version of *Cp*FatB1 with identical genetic architecture and constitutive Cre recombinase expression (Fig. 6d). With OptoCre-*tetA*, octanoic acid production increased 150% over light induction alone when it was supplemented with 1.5 μg/mL tetracycline (Fig. 6e). Higher tetracycline concentrations improved yield further, reaching a 300% increase over light alone when 6 μg/mL tetracycline was added. These concentrations are on the upper range of the resistance levels we see in our MIC experiments for the respective strains, possibly due to the higher OD (OD600 ≈ 0.6) at the time antibiotics are added in our bioproduction protocol. The induction system with OptoCre-*tetA* did not boost production over constitutive expression of *Cp*FatB1, though the addition of antibiotic did greatly improve yield over light alone (Fig. 6e). Importantly, the constitutively expressed versions of both the *cat* and *tetA* constructs showed poor growth profiles compared to the light inducible versions (Supplementary Fig. 6). This growth deficit in the constitutive constructs is problematic, as it may lead to escape mutants and reduce the stability of production strains. Thus, coupling octanoic acid production with resistance selection leads to higher yields of octanoic acid than light induction alone, without the taxing growth deficit associated with continuous production.

Comparing the OptoCre-*cat* and OptoCre-*tetA* constructs, we found that while OptoCre-*cat* showed higher production overall, OptoCre-*tetA* showed a greater increase in production with increasing antibiotic concentrations over the range tested. This could be due to the difference between resistance types (enzymatic vs. efflux) and their impact on population dynamics, or the different promoters of the constructs (P vs. P$_{tet}$), suggesting that these constructs and their induction could likely be optimized further to increase production. This work shows the potential of using light to control antibiotic resistance for metabolic engineering and bioproduction applications.

## Discussion

We have developed and optimized optogenetically controlled systems for four antibiotic resistance genes. Using a blue light-inducible Cre recombinase, we have shown activation of *bla*, *knt*, *cat*, and *tetA* to induce resistance over a range of antibiotic concentrations. These resistance genes span multiple mechanisms and represent antibiotics and resistance genes commonly used in synthetic biology and microbiology labs. A crucial aspect of designing inducible resistance is the level of expression in the uninduced and induced states, and optimal levels vary dramatically with the strength of the resistance gene. Therefore, we tested promoters, RBSs, and enzyme mutants of varying strength to find constructs that show optimal basal expression and fold changes of resistance activation. To control copy number at the DNA level, we also compared constructs chromosomally-integrated in the *nupG* region and on a p15A medium copy plasmid, which improved flexibility in experimental design. We found that optimizing resistance induction constructs at the promoter, RBS, enzyme, and copy number levels can be used as a generalizable approach, providing multiple options for altering expression to allow for flexibility in construct design to meet experimental constraints. For example, studies which require a native promoter, specific plasmid origin, or particular resistance protein to be compatible with other elements of the strain design, can be accommodated as this system is adapted to different use cases or other resistance genes. This system also allows for the expansion of resistance genes beyond the ones shown here, using the optimization approaches mentioned above. Future experiments characterizing expression directly could also be an effective route for further optimizing designs. Using techniques like qPCR or Western

antibiotic resistance can allow for selective growth of cells actively producing both enzymes, thereby preventing the growth of non-expressing cells (Fig. 6c). We used OptoCre-*cat* with the *cat*$_{T172A}$ mutant and OptoCre-*tetA* for selection due to their wide induction ranges and minimal basal resistances on the p15A plasmid backbone. For fatty acid production, we used the highly active variant *Cp*FatB1.2-M4-287 which has been previously shown to boost octanoic acid production[63]. To optimize expression of the *Cp*FatB1 gene without disrupting the resistance activation architecture, we placed it immediately downstream of the resistance gene under expression of a strong computationally-designed RBS denoted R'. Consistent with the protocol for inducing antibiotic resistance alone, we performed induction by exposing cells to blue light for 2 h early in the growth phase. From a bioproduction perspective, an advantage of the Opto-CreVvd system is its irreversibility, which enables permanent activation of *Cp*FatB1 without the need for continuous illumination. This circumvents issues with dynamic light induction systems, where light penetration can become a concern for dense cell cultures in metabolic engineering contexts[40,64]. We found that light induction alone led to a significant increase in octanoic acid production for both the *cat* and *tetA* systems. We next asked whether introducing antibiotic selection

blots could enable relative measurement of transcription and translation levels, or protein fusions with a fluorophore could enable measurement in cases where the fusion is known not to impact function.

We selected the four antibiotic resistance mechanisms here to be broadly applicable for both synthetic biology and microbiology uses. These antibiotics and resistance genes are often used as plasmid selection markers and in synthetic biology control systems[24,44]. As an example of an application, these constructs have the potential to be applied to single-cell selection in microfluidic systems. Selection of single cells of interest from a microfluidic device is a challenge, and current methods require complex optical traps and valve-based microfluidic devices[9]. Combining light-inducible resistance with a DMD[62] for precise targeting of light would allow for antibiotic selection of a cell line of interest from device outflow, without any chip modification or additional hardware requirements beyond light exposure. Using light also allows integration into computational workflows, potentially facilitating automated screening and selection for phenotypes of interest. The permanent switch would ensure that the selection signal is not lost, allowing progeny of selected bacteria to be harvested for characterization at any point after activation. Additionally, given the prevalence of recombinases in cellular logic[28], this system could be modularly integrated with larger recombinase circuits as a logic output controlling cell survival. For instance, recombinase-controlled resistance genes could be broadly used to regulate cell survival only if certain environmental conditions (e.g. small molecule, light, or temperature) are met[28]. In these cases, it may be worth using a constitutive version of OptoCreVvd2, rather than the IPTG-inducible version used here, to simplify the workflow and minimize chemical induction requirements. As both recombinases and the selected resistance genes are commonly used in synthetic biology, this allows for optogenetic control to be straightforwardly incorporated into existing genetic systems.

Further, we envision these light-inducible antibiotic resistance genes being useful as a synthetic system for studying horizontal gene transfer. These systems are well-suited to examine how single-cell resistance gene acquisition events lead to population expansion or decline. Being able to permanently activate resistance gene "acquisition" using light removes the need to rely on observations of infrequent and stochastic natural horizontal gene transfer events. Notably, *bla* is particularly problematic in clinical settings[31], and known to be transferred extensively when the gut is exposed to ampicillin[12]. Recent studies looking at single-cell instances of horizontal gene transfer have also shown that quorum sensing and biofilm structures can be important in initiating transfer events[59,60], however experiments thus far have been limited to the study of infrequent horizontal transfer events. Critically, because not all resistance acquisition events lead to the proliferation of resistant populations[61], synthetic control of resistance acquisition could reveal when and how cells acquiring resistance proliferate. In the future, this system could also be modified to enable resistance to be turned off or controlled reversibly, allowing for both resistance acquisition and loss studies. Using light as an inducer also enables studies with spatial dynamics, beyond what could be achieved using chemical induction. However, light penetration may become an issue for complex 3D spatial geometries. Overall, the spatial and temporal control afforded by these optogenetic systems could allow researchers to determine what cellular arrangements and antibiotic treatment schedules lead to expansion or collapse of microbial communities consisting of resistant cells and their susceptible neighbors[61,65].

This system also provides novel benefits to synthetic bioproduction systems, which we demonstrated in a pilot study using *Cp*FatB1 to increase production of octanoic acid. Although efficient, the expression and enzymatic activity of *Cp*FatB1 are taxing at high levels, resulting in a growth advantage for non-producers and potentially giving rise to cheaters[63,66]. By coupling expression of a resistance gene with the bioproduction enzyme, we can limit growth to only cells expressing *Cp*FatB1, further increasing yield over the equivalent system with light induction alone. This work can be expanded to other bioproduction enzymes of interest, or co-expressed with multiple genes to require expression of several enzymes in a pathway for growth.

Antibiotic resistance genes are ubiquitous and fundamental tools of synthetic biology. However, there are few current methods that allow dynamic regulation of their expression. The ability to activate resistance using light expands this core tool, improving spatiotemporal control and enabling new applications in cellular selection and the study of antibiotic resistance acquisition.

## Methods
### Strains and plasmids
All antibiotic resistance assays use *E. coli* strain MG1655. We constructed plasmids using the Gibson assembly method[67], or using Golden Gate assembly[68] in cases where the constructs contained fragments under 100 base pairs in length (Supplementary Table 1).

Chromosomally integrated constructs were inserted using the Lambda Red recombinase system[1] downstream of *nupG* (forward homology site: GGTTCTGGCCTTCGCGTTCATGGCGATGTTCAAATAT AAACACGT, reverse: GGCGTGAAACGGTTGTACGGTTATGTGTTGAAG TAAGAATAA). Antibiotic resistance cassettes flanked with FRT sites were used to select successful integrations (we used *knt* for *bla* and *cat* activation constructs, *cat* for *knt* and *tetA* activation constructs); these cassettes were then cured using a plasmid-based FLP recombinase on a temperature sensitive origin following the protocol from Datsenko and Wanner[1]. Finally, we cured the temperature sensitive plasmid before subsequent experiments.

The plasmids used for OptoCreVvd2 expression were derived from Sheets et al.[26]. For *bla* activation studies, we changed the plasmid selection cassette from *bla* to *cat*, using the gene from the BglBrick plasmids[44] and primers listed in Supplementary Table 1. The antibiotic resistance activation plasmids were made by changing the respective promoter, RBS, and reporter gene of pBbAk-W4-loxTTlox-mRFP1 from Sheets et al.[26]. Plasmid origins of replication and sequences for *bla*, *knt*, and *cat* were taken from the BglBrick plasmid series[44]. The $cat_{T172A}$ mutation was taken from Ciechonska et al.[51]. The sequence for *tetA* was obtained from AddGene plasmid #74110 (pRGD-TcR) deposited by Hans-Martin Fischer[69]. The sequence for *Cp*FatB1.2-M4-287 was obtained from Hernández Lozada et al. and synthetized using Twist Bioscience[63]. Constitutive *Cp*FatB1.2-M4-287 expression strains were created by co-transforming the same reporters used for the light expression experiments with pBbE5a-Cre, which acts constitutively. All plasmid-based resistance constructs contain the p15A origin. Plasmid-based constructs for *knt* activation contain *cat* as a selection marker, and constructs for *bla*, *cat*, and *tetA* activation contain *knt* as a plasmid selection marker. Positive controls for *bla* and *tetA* contain the respective gene with its native promoter and RBS on a plasmid with the p15A origin. Positive controls for *knt* and *cat* contain the respective gene with its native promoter and RBS on a plasmid with the p15A origin for studies using a plasmid-based reporter, or integrated into the *nupG* region for studies using chromosomally integrated reporters. Promoters used were medium, medium-low, and low-strength variants of T7 A1[70], denoted P (TTATCAAAAAGAGTA TTGCAT TAAAGTC-TAACCTATAG GAATCT TACAGCCATCGAGAGGGACACGGCGAA), P* (TTATCAAAAAGAGTA TTGTCT TAAAGTCTAACCTATAG GATTCT TACAGCCATCGAGAGGGACACGGCGAA), and P** (TTATCAAAAA-GAGTA TTGTAA TAAAGTCTAACCTATAG GATTTT TACAGCCATCGA-GAGGGACACGGCGAA). Underlines indicate mutations from the original T7 A1 promoter. Ribosome binding site R is the RBS of gene 10 in the T7 phage (TTTAAGAAGGAGATATACAT)[47]. R* (ATCACTC-TACGGCCAGCTGCAAAC) was computationally designed using De Novo DNA version 2.1 to have a 10x weaker translation strength

(14.8 A.U.) compared to R (148 A.U.), and R' (TTTGTTTAATTAC-TAAGCGGGAGGTTAT) was designed for increased translation strength (100,000 A.U.)[48,71]. The rrnB terminator BBa_B0015, used in the original pBbAk-W4-loxTTlox-mRFP1 from Sheets et al.[26], was used as the *loxP*-flanked terminator in all constructs unless otherwise noted. The strong synthetic terminator L3S2P21[53] was synthesized by IDT and cloned into an mCherry variant of the original fluorescent reporter plasmid. Primers used to change each of these elements are included in Supplementary Table 1.

Plasmids and strains from this study and their sequences are available on AddGene (https://www.addgene.org/Mary_Dunlop/).

## Blue light exposure

Strains were grown overnight from a single colony in selective LB media containing 100 µg/mL carbenicillin, 30 µg/mL kanamycin, or 25 µg/mL chloramphenicol as required for plasmid maintenance. Cultures were refreshed 1:100 in selective M9 minimal media (M9 salts supplemented with 2 mM $MgSO_4$, 0.1 mM $CaCl_2$, and 0.1% glucose) for 2 h with 100 µM IPTG for induction of OptoCreVvd2 split recombinase production. Cultures were then either exposed to blue light or kept in the dark for 2 h. Light exposure was performed using a 24-well light plate apparatus (LPA)[72] using 1 mL cultures with two 465 nM wavelength LEDs per well (ThorLabs LED465E), with a total output of 120 µW/cm² per well. For the light intensity variation experiment, the LPA was used to deliver 5, 60, or 120 µW/cm² for 0.5, 1, or 2 h following the same protocol.

## Minimum inhibitory concentration measurement

Minimum inhibitory concentrations (MICs) were measured based on the protocol outlined in Wiegand et al.[73]. Antibiotic stocks were made by dissolving the antibiotic in sterile distilled water (carbenicillin, kanamycin, tetracycline) or 99% ethanol (chloramphenicol), with concentrations normalized for potency based on CLSI standards[74]. Assay plates for measuring the MIC were prepared by performing serial dilutions of antibiotic in 100 µL M9 minimal media in 96-well plates. Low glucose media was used to reduce growth variability by creating carbon-limiting, rather than nutrient-limiting, growth conditions[45]. Antibiotic concentrations were selected to include values that spanned the MIC levels for the dark and light state cultures in each experiment. Immediately following light exposure, cultures were normalized by dilution to the lowest optical density (OD) of each experiment. Normalized cultures were then diluted 1/25 into 96-well plates in triplicate and grown overnight for 18 h at 37 °C. The OD absorbance reading of each well was then measured at 600 nm (OD600) using a BioTek Synergy H1 plate reader. Post-sample sequencing for OptoCre-*cat* reporter strain alone was done on liquid culture from the 0 µg/mL and 75 µg/mL (just below MIC) growth conditions, at the end of the 18 h growth period for samples shown in Supplementary Fig. 2. Samples were amplified using Phusion polymerase by forward (AAGCCATC-CAGTTTACTTTG) and reverse (CCAGCTGAACGGTCTGGTTATAGG) primers to encompass the terminator region.

## Colony forming unit measurement

Colony forming units (CFU) were measured following the micro-spotting protocol outlined in Sieuwerts et al.[75]. After MIC plates were grown overnight for 18 h, cultures were serially diluted 1:10 in 1x M9 salts in a 96-well plate. Dilutions from $10^{-1}$ to $10^{-6}$ were spot-plated, with 5 µL on LB-agar plates in duplicate for each well ($n = 6$ for each condition). Plates were grown overnight at 37 °C and colonies were counted by hand the next day for the lowest dilution with countable colonies for each sample. CFU counts per mL were then calculated by multiplying dilution level by the average number of colonies counted by per condition, normalizing for the 5 µL volume plated.

## Microscopy and image analysis

Strains were grown overnight from a single colony in selective LB media. Cultures were refreshed 1:100 in selective M9 minimal media for 2 h with 100 µM IPTG for induction of OptoCreVvd2. Samples were then placed on 1.5% low melting agarose pads made with M9 minimal media containing 100 µM IPTG and 400 µg/mL kanamycin (*knt* activation) or 60 µg/mL chloramphenicol (*cat* activation). Samples were grown at 30 °C to prevent pads from drying out, and imaged every 15 min (*knt* activation) or 20 min (*cat*$_{T172A}$ activation) for 18 h. For whole-frame illumination experiments, cells were imaged at 100x using a Nikon Ti-E microscope. Blue light exposure was provided by a LED ring (Adafruit NeoPixel 1586), which was fixed above the microscope stage and controlled by an Arduino with a custom Matlab script for a total output of 330 µW/cm². Images were segmented and analyzed using the DeLTA 2.0 software[76]. Cell divisions were annotated by hand. For digital micromirror device (DMD) experiments illuminating half of the field of view, cells were imaged at 100x using a Nikon Ti-2 microscope. Light exposure was provided by a DMD (Mightex Polygon 400) connected to the illumination light path of the Nikon Ti-2 chassis. Light from the DMD was passed through a 10% neutral density filter (Chroma UVND 1.0) for a total of 1.2% power, or 160 mW/cm², for 4 out of 5 min for each imaging cycle. An Arduino Uno microcontroller was used to synchronize the camera, the illumination source, and the DMD for image acquisition[77].

## Fatty acid production and measurement

Strains were grown overnight from a single colony in selective LB media. Cultures were refreshed 1:50 in selective M9 minimal media containing 2% glucose[78] and 100 µM IPTG for strains with Opto-CreVvd2. Cultures were grown to an OD600 ≈ 0.2. Induced cultures containing OptoCreVvd2 were then exposed to blue light for 2 h. Immediately following light exposure, 150, 300, or 600 µg/mL chloramphenicol or 1.5, 3, or 6 µg/mL tetracycline was added to OptoCreVvd2 induced cells. Following 20 h of growth post light induction, 400 µL of the cultures were taken and prepared for gas chromatography-mass spectrometry (GC-MS) quantification. Constitutively expressed *Cp*FatB1 strains were grown in identical conditions but did not receive light or antibiotic. Fatty acid extraction and derivatization into fatty acid methyl esters was completed as described by Sarria et al.[79]. An internal standard of nonanoic acid (C9) was added to the sample at a final concentration of 88.8 mg/L and vortexed prior to extraction. The samples were analyzed with an Agilent 6890 N/Agilent 5973 MS detector using a DB-5MS column. The inlet temperature was set to 300 °C with flow at 4 mL/min. The oven heating program was initially set to 70 °C for 1 min, followed by a ramp to 290 °C at 30 °C/min, and a final hold at 290 °C for 1 min. The nonanoic internal standard was used for quantification of octanoic acid titer.

## Statistical analysis

OD600 values in MIC curves and bar plots are reported as the mean of three samples ± the standard deviation. Colony count values for CFU measurements are reported as the mean of six samples consisting of two dilutions and platings for each MIC data point. Fatty acid production measured using GC-MS is reported as the mean of three biological replicates for each condition ± the standard deviation. Statistical significance (*P* value) for fatty acid production was assessed using a two-tailed Welch's *t*-test.

## Reporting summary

Further information on research design is available in the Nature Portfolio Reporting Summary linked to this article.

## Data availability

Data needed to evaluate the conclusions are present in the manuscript and/or the Supplementary Materials. Source data are provided with this paper.

## Code availability

Image analysis code is based on the DeLTA 2.0 algorithm available at https://gitlab.com/dunloplab/delta.

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

## Acknowledgements

We thank Heidi Klumpe for her helpful comments on the manuscript. We thank Caroline Blassick for creating the $cat_{T172A}$ mutant and Mark Isalan for pointing us to this variant. This work was supported by NIH grant R01AI102922 and DOE grant DE-SC0019387. M.B.S. received support through the NIH training grant T32 EB006359.

## Author contributions

M.B.S. and M.J.D. conceived and designed the experiments. M.B.S. performed resistance induction experiments and analyzed the data. M.B.S. and N.T. performed fatty acid production experiments and analyzed the data. M.B.S. and M.J.D. wrote the manuscript with input from N.T.

## Competing interests

The authors declare no competing interests.
