## [Peer Review File · Nature Communications]

Reviewers' Comments:

Reviewer #1:

Remarks to the Author:

In this article, the authors describe light-inducible control of antibiotic resistance in *Escherichia coli*. To achieve this control, the authors use a blue-light sensitive split CRE-recombinase (OptoCreVvd2) to lox-out a terminator blocking transcription of an antibiotic resistance gene. In this way, blue light causes removal of the terminator and the subsequent constitutive expression of the antibiotic resistance gene. The OptoCreVvd2 system was developed, optimized, and characterized by the same authors in a previous work (Sheets, et al 2020 ACS Synthetic Biology) so the innovation here is using this system to control four different resistance genes (bla-carbenicillin, knt-kanamycin, cat-chloramphenicol, tetA-tetracycline). The authors optimize resistance induction using standard synthetic biology/genetic engineering techniques including varying promoters, ribosome bindings sites (RBS) and the enzyme variant strength for both chromosome and plasmid-based constructs. The authors also demonstrate that linking inducible resistance to expression of an enzyme required for heterologous fatty acid production can increase production of octanoic acid.

This is an interesting technique for spatiotemporal control of antibiotic resistance in *Escherichia coli* which could be utilized to understand antibiotic resistance dynamics in microbial communities as well as for biotechnology applications. Gutierrez Mena, et al 2022 Nature Communications recently demonstrated a complimentary approach to light-inducible antibiotic resistance using a light-controllable T7 polymerase. Koganezawa (Koganezawa, et al 2022 Elife) utilized an optogenetic Cre-lox system (based around a different set of light-dimerizing proteins than utilized in OptoCreVvd2) to induce antibiotic susceptibility by loxing out antibiotic resistance genes in response to light and followed the life history of these suddenly susceptible cells to understand how cells adapt to lethal genetic modifications through physiological resistance. The current manuscript complements these studies by characterizing and optimizing light-inducible resistance to more kinds of antibiotics and demonstrates a small proof of principle for biotechnology applications.

Overall, the paper is well-written and the experiments appear technically sound. There are some major points that should be addressed prior to publication:

Major Comments:

- While control of antibiotic resistance in *Escherichia coli* is the focus of this article (and has been demonstrated by Gutierrez Mena et al, Olendorf, et al and Koganezawa, et al as the authors point out), optogenetic antibiotic resistance has also already been demonstrated in other microbial systems including *S. cerevisiae* (Moreno Morales, et al 2021—Nourseothricin) and *Y. lipolytica* (Wang, et al 2022 Int. J. Mol Sci—Bleomycin). This is worth pointing out in the text with appropriate citations to situate the authors' work in the context of the larger fields of optogenetics and microbial engineering.

- Line 90/Line 224: Given that basal expression is one of the limiting factors in system function, did the authors think about exploring different strength bacterial terminators (e.g. Hudson and Wieden, 2019 Syn Bio and others). Why were gene copy, promoter, RBS, and coding sequence more appropriate choices for system optimization?

- Line 146: The statement that the OptoCreVvd2 system can "precisely induce" beta-lactamase resistance seems like an overstatement given the leakiness of the CFU measurements in Figure 1d. Indeed, one problem seen throughout the paper is that simply introducing the resistance cassette (i.e. compare light gray WT to light gray CTRL -) in many cases provides a significant amount of resistance. Indeed, comparing the negative control and dark samples (i.e. compare dark gray and black lines) shows that introduction of the resistance gene is often the dominant effect causing basal resistance, ie, CRE recombinase doesn't need to be present at any level for this effect to happen. This relates to another comment regarding whether or not exploring different terminators would be a relevant optimization. In addition, is this resistance due to basal expression of the resistance gene due to terminator read-through, spontaneous loxing/recombination, or mutation that knocks out the terminator? These would lead to different population effects, generating populations with weak resistance (due to weak basal expression) vs initiating small populations with full resistance (terminator knockouts). This would be a useful characterization of the system.

- Line 198: It is a stretch to say that the added functionality provided by plasmids increases the "ease" at which these constructs can be used. In fact, looking at the data in Figure 2d, introducing

the resistance gene on a plasmid in many cases destroys the ability to differentiate between the negative control, the dark sample, and light (i.e. compare dark gray, black, and dark blue lines). This effect is even more severe for the bla resistance gene (see Figure S1) where introducing resistance on the p15A plasmid completely eliminates differences between the negative control, dark, and light samples.

- Throughout the manuscript the connection between expression levels of the resistance gene and the survival of cells is indirect. We don't know for each resistance gene how much is being expressed, or directly how the different optimization techniques change protein levels. Some of this was done in Sheets, et al 2020 but, for example, I don't think that chromosomal integration of the RFP reporter was ever done in that paper. It would be nice to be able to connect the successes or difficulties in tuning resistance to protein expression. I also don't have a good sense for how homogeneous induction of resistance is in response to blue light. Should we expect antibiotic resistance to be tunable with light intensity? Or is resistance about having a threshold illumination level and then waiting long enough for cells to lox out the terminator and start expressing resistance? And are differences due to fraction of the population that loxes out the reporter, duration after lox out, or both?

- Figure 5 would be more relevant and exciting if it demonstrated spatial control. This seems like a straightforward experiment using a DMD with the agar pad, unless there is some reason (i.e. diffusion) that it actually doesn't work well. Which would be useful to know.

- The requirement for IPTG induction (to induce OptoCreVvd2) is only mentioned in the methods. This is actually a serious constraint on the system, as it means that to use the system chemical induction is always required (which has implications for both experiments for basic research as well as bioproduction applications) and it also provides an extra layer of reduction in terms of basal expression (because Cre recombinase is not "hanging around" until just before light induction) that is important to discuss and make crystal clear to the reader. For example, if you induced with IPTG for 4 hours or 8 hours would you start to see lots more basal loxing? I suspect so.

- The experiment coupling antibiotic resistance to octanoic acid production is interesting and could hint at biotechnology applications, but I have some concerns. First, is light induction (rather than light-off) a feasible strategy in bioproduction given concerns with light penetration in culture? Second, unless I am misreading the data, an important control wasn't done. Namely, a strain with constitutive expression of CpFatB1. As is, it isn't clear that the light inducible strategy is an improvement over constitutive expression. And indeed, for Figure 6d, light + drug minimally improves production. It is also worth delving further into why production is being improved. Are sensitive cells really being killed off to free up more resources? Can this be shown directly? Not knowing the "whys" is related to another concern—namely, the difference between chloramphenicol and tetracycline resistance in regulating octanoic acid production. The differences in the dark controls are presumably due to different basal resistance (?). The differences between the induced conditions with different amounts of antibiotic could be due to the difference in mechanism between the two resistance types, but it could be that chloramphenicol would also show that increased gradient at lower concentrations (we just didn't see the dynamic range in what was tested). Presumably tetracycline would also level out at some point?

- Related to the octanoic acid experiment, it is difficult to compare to the previous antibiotic resistance experiments because the ODs and glucose concentrations are so different between these experiments (e.g. compare Fig. 6 to Fig. 2 and Fig. 4). Why use such different culture conditions (0.1% glucose vs 2% glucose), making these experiments difficult to compare?

- Line 422, line 465: Generally, it isn't clear why certain culture conditions were picked for some experiments and not others. For Fig. 1,2,3 & 4 the ODs are quite low presumably because experiments were done in 0.1% glucose. Why was this choice made? Later, for the bioproduction experiment, cells were grown in 2% glucose. Please explain these different choices and their effect on the results.

- One of the benefits of light is that it can be controlled in space, in fact, one could argue that this is the primary advantage over chemical inducers. Unfortunately, the authors don't demonstrate spatial control in this article, which would be important for subsequent use in determining how structure in microbial communities interacts with drug resistance to determine population outcomes. At least the authors should expand the discussion of the limitations on spatial control. For example, there are limitations due to light penetration (i.e. experiments probably need to be done in monolayers). In addition, there are limits set by diffusion of the resistance mechanism itself (i.e. beta-lactamase)

- For optogenetic approaches to be useful for biotechnology applications, particularly use in larger

scale bioreactors, light-off control has been suggested to be a better approach (i.e. your gene of interest is induced by the absence of light, as would happen as a culture becomes more dense and light penetration falls—see for example Lalwani, et al 2021; Zhao, et al 2018). That is not the approach taken here, so the authors should discuss the utility of their optogenetic coupling of antibiotic resistance to a key metabolic enzyme for increasing bioproduction and the potential for scale-up.

Minor Comments:

- The manuscript emphasizes the importance and utility of dynamic antibiotic resistance control to understand and control microbial populations. However, the control developed in the paper is not dynamic in the sense that it is not reversible. It is a temporally controlled “on” switch. This distinction should be made clear in the text.
- Lines 54-57: The phrasing here describing the Gutierrez Mena, et al paper is somewhat misleading and in our minds potentially not emphasizing the strengths of the current manuscript. The Gutierrez Mena approach (light-controlled T7 polymerase to induce an antibiotic resistance gene) is almost certainly generalizable (to some extent) although this wasn’t tested. So what is novel in the current manuscript is both that generalizability was tested (and indeed four important different resistance mechanisms compared and contrasted) and the Sheets, et al system is irreversible. For some applications this is a disadvantage, but for many others (e.g. one-time induction before production cultures become too saturated, spatial applications, light-sensitive strains) it is an advantage. The language in these sentences could be slightly tweaked to appropriately convey these nuances.
- Line 66: “minimal basal expression of activated genes” is almost an oxymoron (i.e. if they are activated, it is hopefully not minimally). Would “minimal basal expression in uninduced cells” be more appropriate?
- Line 329: “digital micromirror device” is not defined and there is no reference. It might be that at this point this technology is familiar enough to most groups, but maybe not. It is also true that there are other technologies that could potentially be used for spatial illumination and that might be worth mentioning.
- All Figures: At least on our research group’s and department’s printers, the light gray line tends to disappear when the manuscript is printed. It might be worth switching to a darker color.
- Could the authors comment on why light-induced resistance is never able to reach the same levels of resistance as the positive control strains?
- The layout of some figures (i.e. Fig. 2 and Fig. 3) is confusing in terms of the organization of B, C, and D. Would it be more clear to separate these elements into columns or rows? Should the legend in the graphs in C also be repeated in the graphs in D?
- One potential downside of this system is that it is not reversible, but it would be possible to use recombinases to develop a reversible switch (with more circuitry). The potential benefits and limitations of this and in general the potential for reversibility should be added to the discussion.
- Line 321: What might require those conditions?
- Line 351: Should be “synthetic control of resistance acquisition” rather than gene transfer, as gene transfer wasn’t implemented in this paper (although control over HGT would be a very interesting way to explore antibiotic resistance mechanisms and downstream population effects).
- Line 357-359: Do you mean to say here that the enzyme is efficient, but taxing to express at high levels? Or the reaction catalyzed by the enzyme?
- Line 424-426: If these experiments were done in an LPA presumably they were done in 24-well plates. What is the volume of cell culture?
- Line 458-459: Should the intensity of the light at the cells also be included here?

Reviewer #2:

Remarks to the Author:

In the manuscript, the authors coupled antibiotic-resistance genes with optogenetics to engineer bacteria with light-induced antibiotic-selective features. They utilized the well-established optogenetic regulator loxP/split-Cre system which can sense blue light and irreversibly activate gene expression. Four commonly used antibiotics with different killing mechanisms and the corresponding resistance genes were engineered to be light-active. They showed that antibiotic-resistance genes could be successfully induced and thus facilitates survival against the antibiotics. Moreover, by manipulating the plasmid backbones, promoters, and RBS, the dynamic range of the

antibiotics concentrations the bacteria tolerate can be tuned to the desired value. Then they screened the single-cell growth process in time-lapse with or without light induction, aiming to showcase the possibility of single-cell level behavior of antibiotic resistance acquisition after horizontal gene transfer. At last, they further coupled the system in a fermentation strain with the chemical production pathway so that the cheaters could be killed to improve the yield. The results are overall solid and I have no concerns regarding its technical soundness. However, the demonstration of its potential applications could be further developed. Following are my concerns on this paper:

1. In the fermentation application part, the author was trying to demonstrate that their system can improve the yield. However, the authors did not test the original fermentation strain as the control. The improvements of their engineered system therefore are not characterized. In addition, the fermentation process is a relatively long-term process. Using light induction to kill cheaters could face many issues in real world scenarios, i.e. when to induce, how frequent, and impacts on fermentation strain, etc. To claim its application potentials, more validations are needed.
2. The authors propose in the paper that their system to be used as a model to study the spread of antibiotic resistance genes obtained by horizontal gene transfer, based on the assumption that the light-induced resistance gene expression could be regarded as similar to the acquisition of the resistance genes by horizontal gene transfer. But resistance spread is largely through selection or more subsequent horizontal gene transfer, which is different from transient gene expression increase in the engineered strain after light induction. I would recommend the authors to elaborate how the transient gene expression increase could be an important aspect in studying antibiotic resistance spread. This system has big potentials in investigating one important aspect that could be overlooked. But in current form, this connection is not probed with enough depth.
3. The in-text citation numbers should be inside sentences before the punctuations.
4. I suggest adding a few subtitles to the results section to improve the reading experience.

Response to Reviewers

Manuscript #NCOMMS-22-23220A-Z

We appreciate the reviewers' detailed comments and suggestions to improve the quality and clarity of the manuscript. We have addressed all of their concerns in the revised manuscript. A complete list of changes and detailed responses to the comments follows. Highlights include new experimental results:

- New data comparing *CpFatB1* light inducible strains to constitutive controls (Fig. 6, Fig. S6).
- New data characterizing the impact of light intensity and duration on resistance activation (Fig. S2).
- New data comparing the *loxP*-flanked transcription terminator used here to another strong terminator (Fig. S3) and expanded discussion and analysis of basal expression levels.
- New data highlighting the growth recovery of cells from time-course microscopy experiments (Fig. S4).
- New data demonstrating the use of a digital micromirror device (DMD) to activate resistance in specific regions within a field of view (Fig. S5).

Reviewer #1:

In this article, the authors describe light-inducible control of antibiotic resistance in *Escherichia coli*. To achieve this control, the authors use a blue-light sensitive split CRE-recombinase (OptoCreVvd2) to lox-out a terminator blocking transcription of an antibiotic resistance gene. In this way, blue light causes removal of the terminator and the subsequent constitutive expression of the antibiotic resistance gene. The OptoCreVvd2 system was developed, optimized, and characterized by the same authors in a previous work (Sheets, et al 2020 ACS Synthetic Biology) so the innovation here is using this system to control four different resistance genes (bla-carbenicillin, knt-kanamycin, cat-chloramphenicol, tetA-tetracycline). The authors optimize resistance induction using standard synthetic biology/genetic engineering techniques including varying promoters, ribosome bindings sites (RBS) and the enzyme variant strength for both chromosome and plasmid-based constructs. The authors also demonstrate that linking inducible resistance to expression of an enzyme required for heterologous fatty acid production can increase production of octanoic acid.

This is an interesting technique for spatiotemporal control of antibiotic resistance in *Escherichia coli* which could be utilized to understand antibiotic resistance dynamics in microbial communities as well as for biotechnology applications. Gutierrez Mena, et al 2022 Nature Communications recently demonstrated a complimentary approach to light-inducible antibiotic resistance using a light-controllable T7 polymerase. Koganezawa (Koganezawa, et al 2022 Elife) utilized an optogenetic Cre-lox system (based around a different set of light-dimerizing proteins than utilized in OptoCreVvd2) to induce antibiotic susceptibility by loxing out antibiotic resistance genes in response to light and followed the life history of these suddenly susceptible cells to understand how cells adapt to lethal genetic modifications through physiological resistance. The current manuscript complements these studies by characterizing and optimizing light-inducible resistance to more kinds of antibiotics and demonstrates a small proof of principle for biotechnology applications.

Overall, the paper is well-written and the experiments appear technically sound. There are some major points that should be addressed prior to publication:

These are very comprehensive and helpful comments, and we appreciate the extent to which the reviewer understands the current state of the field and sees how the work presented in this manuscript complements other related studies. Thank you!

Major Comments:

- While control of antibiotic resistance in *Escherichia coli* is the focus of this article (and has been demonstrated by Gutierrez Mena et al, Olendorf, et al and Koganezawa, et al as the authors point out), optogenetic antibiotic resistance has also already been demonstrated in other microbial systems including *S. cerevisiae* (Moreno Morales, et al 2021—Nourseothricin) and *Y. lipolytica* (Wang, et al 2022 Int. J. Mol Sci—Bleomycin). This is worth pointing out in the text with appropriate citations to situate the authors' work in the context of the larger fields of optogenetics and microbial engineering.

We thank the reviewer for bringing these publications to our attention, and have added this broader context to the Introduction.

→ See Introduction

- Line 90/Line 224: Given that basal expression is one of the limiting factors in system function, did the authors think about exploring different strength bacterial terminators (e.g. Hudson and Wieden, 2019 Syn Bio and others). Why were gene copy, promoter, RBS, and coding sequence more appropriate choices for system optimization?

To address this question, we selected a strong synthetic terminator, L3S2P21, from the library in (Chen *et al.* 2013). We choose this particular terminator because it was the strongest of all 582 characterized terminators in that study. However, we found that basal gene expression levels were comparable between our initial strong terminator design and the new terminator, leading us to focus on other optimization routes. However, we note this as a potential approach for reducing basal expression and include references to bacterial terminator papers and other design strategies. Data comparing designs with the original terminator and L3S2P21 are shown in Fig. S3.

→ See Fig. S3

→ See Results / Protein-level optimization of chloramphenicol resistance section

- Line 146: The statement that the OptoCreVvd2 system can “precisely induce” beta-lactamase resistance seems like an overstatement given the leakiness of the CFU measurements in Figure 1d. Indeed, one problem seen throughout the paper is that simply introducing the resistance cassette (i.e. compare light gray WT to light gray CTRL -) in many cases provides a significant amount of resistance. Indeed, comparing the negative control and dark samples (i.e. compare dark gray and black lines) shows that introduction of the resistance gene is often the dominant effect causing basal resistance, ie, CRE recombinase doesn't need to be present at any level for this effect to happen. This relates to another comment regarding whether or not exploring different terminators would be a relevant optimization. In addition, is this resistance due to basal expression of the resistance gene due to terminator read-through, spontaneous loxing/recombination, or mutation that knocks out the terminator? These would lead to different

population effects, generating populations with weak resistance (due to weak basal expression) vs initiating small populations with full resistance (terminator knockouts). This would be a useful characterization of the system.

To check whether leakiness is due to spontaneous loxing or mutations impacting the terminator, as opposed to transcriptional readthrough of the terminator, we sequenced the (-) Control culture of OptoCre-*cat* with no Cre protein after the light induction experiment in Fig. S2. We sequenced the culture from the no chloramphenicol and 75 µg/mL chloramphenicol conditions, which was the highest condition exhibiting growth, and found no evidence of mutations. These results suggest that transcriptional readthrough is the likely cause of the leaky expression observed in the (-) Control. In our previous paper (Sheets *et al.* 2020) we characterized this basal effect and also found no evidence of spontaneous loxing. Additionally, we see this basal effect consistently between experiments and across genes, which suggests a more continuous process such as terminator readthrough compared to rarer events such as mutations that inactivate a terminator. However, we now note these possibilities in the manuscript and include a discussion of the sequencing results.

→ See Results / Protein-level optimization of chloramphenicol resistance section

→ See Methods

The reviewer's point that "precisely induce" may be an overstatement is well taken. We have changed the text to simply say "induce".

→ See Results / Light induction of beta-lactamase resistance section

- Line 198: It is a stretch to say that the added functionality provided by plasmids increases the "ease" at which these constructs can be used. In fact, looking at the data in Figure 2d, introducing the resistance gene on a plasmid in many cases destroys the ability to differentiate between the negative control, the dark sample, and light (i.e. compare dark gray, black, and dark blue lines). This effect is even more severe for the *bla* resistance gene (see Figure S1) where introducing resistance on the p15A plasmid completely eliminates differences between the negative control, dark, and light samples.

The use of "ease" was intended to refer to removing the need to integrate resistance constructs into the chromosome (it is easier to transform a plasmid than integrate a construct chromosomally). We have clarified this in text to avoid future confusion.

→ See Results / Optimization of kanamycin resistance section

- Throughout the manuscript the connection between expression levels of the resistance gene and the survival of cells is indirect. We don't know for each resistance gene how much is being expressed, or directly how the different optimization techniques change protein levels. Some of this was done in Sheets, et al 2020 but, for example, I don't think that chromosomal integration of the RFP reporter was ever done in that paper. It would be nice to be able to connect the successes or difficulties in tuning resistance to protein expression.

This is a good point, and the reviewer is correct that Sheets et al. 2020 did not perform chromosomal insertion characterizations. Here, we were wary of inferring too much about expression levels from fluorophore results because they may not translate directly to expression

of different resistance proteins, e.g. due to different folding times, and limits of fluorescence detection. We elected to focus on optimizing survival here, however we now note in the Discussion that experiments characterizing expression could be a fruitful route for further optimizing designs.

→ See Discussion

I also don't have a good sense for how homogeneous induction of resistance is in response to blue light.

To better understand induction homogeneity, we used data from single-cell microscopy and quantified the number of individual cells present at the start of the movie that begin to recover (Fig. S4). We define recovery as experiencing at least one additional division event. In addition to including the new results, we also discuss some additional features of the data, e.g. cells in the dark can occasionally divide once, but they rarely divide again.

→ See Fig. S4

→ See Results / Single-cell microscopy showing resistance activation section

Should we expect antibiotic resistance to be tunable with light intensity? Or is resistance about having a threshold illumination level and then waiting long enough for cells to lox out the terminator and start expressing resistance? And are differences due to fraction of the population that loxes out the reporter, duration after lox out, or both?

To answer this, we performed a light tuning experiment to determine the impact of both light intensity and timing on survival, and have included this as Fig. S2. The question about fraction of the population is addressed with the new data in Fig. S4.

→ Fig. S2

→ Fig. S4

→ See Results / Protein-level optimization of chloramphenicol resistance section

- Figure 5 would be more relevant and exciting if it demonstrated spatial control. This seems like a straightforward experiment using a DMD with the agar pad, unless there is some reason (i.e. diffusion) that it actually doesn't work well. Which would be useful to know.

We agree that spatial control and DMD experiments are an exciting avenue for this work. As an initial proof-of-concept, we have added data where we use a DMD to activate resistance (Fig. S5). One notable difference between protocols is that a DMD is designed to deliver intense periods of light, compared to the lower intensity exposures we used for the system so far. Although light diffusion appears to cause some spurious activation events, we are hopeful that future optimization of DMD activation protocols can help address this and the initial DMD results are promising.

→ See Fig. S5

→ See Results / Single-cell microscopy showing resistance activation section

- The requirement for IPTG induction (to induce OptoCreVvd2) is only mentioned in the methods. This is actually a serious constraint on the system, as it means that to use the system chemical induction is always required (which has implications for both experiments for basic

research as well as bioproduction applications) and it also provides an extra layer of reduction in terms of basal expression (because Cre recombinase is not “hanging around” until just before light induction) that is important to discuss and make crystal clear to the reader. For example, if you induced with IPTG for 4 hours or 8 hours would you start to see lots more basal loxing? I suspect so.

The use of IPTG induction here is to make handling of strains prior to light induction experiments (for transformations, overnights) easier. The system is not restricted to chemical induction and could be used with a constitutive OptoCreVvd2, however this would require careful handling to keep it in the dark, which was not realistic in our laboratory setting. We have added discussion on this point that references our prior results in Sheets et al. 2020 showing that IPTG induction for 10 hours in the dark did not lead to any basal loxing.

→ See Results / Protein-level optimization of chloramphenicol resistance

- The experiment coupling antibiotic resistance to octanoic acid production is interesting and could hint at biotechnology applications, but I have some concerns. First, is light induction (rather than light-off) a feasible strategy in bioproduction given concerns with light penetration in culture?

To clarify, because the OptoCre system is irreversible, cells only need light at some point during culture and not continuously for production. Thus, light exposure could be performed early to allow light penetration, or for a longer period in a well-mixed culture to ensure a sufficient amount of cells are activated. We have updated the text to highlight this.

→ See Results / Improving octanoic acid production using antibiotic resistance selection section

Second, unless I am misreading the data, an important control wasn't done. Namely, a strain with constitutive expression of CpFatB1. As is, it isn't clear that the light inducible strategy is an improvement over constitutive expression. And indeed, for Figure 6d, light + drug minimally improves production.

We thank the reviewer for raising this point and note that Reviewer #2 raised similar concerns. To address this, we added a constitutive expression control for the CpFatB1 experiments (Fig. 6d-e). For OptoCre-*cat*, we found that our inducible system improved yields over constitutive expression. For OptoCre-*tetA*, our inducible system with antibiotic had similar yield to the constitutive version. However, both of the constitutive versions showed poor growth profiles (Fig. S6). This is a critical issue for the constitutive design because burden of this level will likely lead to escape mutants, reducing stability of the production system.

→ See Fig. 6d-e

→ See Fig. S6

→ See Results / Improving octanoic acid production using antibiotic resistance selection section

It is also worth delving further into why production is being improved. Are sensitive cells really being killed off to free up more resources? Can this be shown directly? Not knowing the “whys” is related to another concern—namely, the difference between chloramphenicol and tetracycline resistance in regulating octanoic acid production. The differences in the dark controls are presumably due to different basal resistance (?). The differences between the induced conditions

with different amounts of antibiotic could be due to the difference in mechanism between the two resistance types, but it could be that chloramphenicol would also show that increased gradient at lower concentrations (we just didn't see the dynamic range in what was tested). Presumably tetracycline would also level out at some point?

We agree these are interesting questions and are important for the implementation of the fatty acid production system. As this work focuses on the implementation of the light inducible resistance and uses fatty acid production as a proof-of-concept, we have added these possible future directions to the Discussion.

→ See Discussion

- Related to the octanoic acid experiment, it is difficult to compare to the previous antibiotic resistance experiments because the ODs and glucose concentrations are so different between these experiments (e.g. compare Fig. 6 to Fig. 2 and Fig. 4). Why use such different culture conditions (0.1% glucose vs 2% glucose), making these experiments difficult to compare?

- Line 422, line 465: Generally, it isn't clear why certain culture conditions were picked for some experiments and not others. For Fig. 1,2,3 & 4 the ODs are quite low presumably because experiments were done in 0.1% glucose. Why was this choice made? Later, for the bioproduction experiment, cells were grown in 2% glucose. Please explain these different choices and their effect on the results.

We followed field-specific protocols, which are different for the two applications. For the resistance characterization experiments we used growth conditions based on Andreani *et al.* 2021. For the octanoic acid experiments we used a standard media for fatty acid bioproduction (Zhang *et al.* 2012), which includes higher glucose to allow for a longer production period. We have also clarified this with citations in-text.

→ See Methods / Minimum inhibitory concentration measurement

→ See Methods / Fatty acid production and measurement

- One of the benefits of light is that it can be controlled in space, in fact, one could argue that this is the primary advantage over chemical inducers. Unfortunately, the authors don't demonstrate spatial control in this article, which would be important for subsequent use in determining how structure in microbial communities interacts with drug resistance to determine population outcomes. At least the authors should expand the discussion of the limitations on spatial control. For example, there are limitations due to light penetration (i.e. experiments probably need to be done in monolayers). In addition, there are limits set by diffusion of the resistance mechanism itself (i.e. beta-lactamase)

As described above, we have included an experiment using a DMD for structured spatial illumination (Fig. S5). We have also added new text surrounding these possible uses and limitations in the Discussion.

→ See Fig. S5

→ See Discussion

- For optogenetic approaches to be useful for biotechnology applications, particularly use in larger scale bioreactors, light-off control has been suggested to be a better approach (i.e. your

gene of interest is induced by the absence of light, as would happen as a culture becomes more dense and light penetration falls—see for example Lalwani, et al 2021; Zhao, et al 2018). That is not the approach taken here, so the authors should discuss the utility of their optogenetic coupling of antibiotic resistance to a key metabolic enzyme for increasing bioproduction and the potential for scale-up.

As mentioned in a previous comment, light induction here may be feasible given the irreversible nature of the OptoCreVvd system. We have updated the text to clarify and highlight this. We have also verified that the Lalwani and Zhao papers have been referenced as background.
→ See Results / Improving octanoic acid production using antibiotic resistance selection

Minor Comments:

- The manuscript emphasizes the importance and utility of dynamic antibiotic resistance control to understand and control microbial populations. However, the control developed in the paper is not dynamic in the sense that it is not reversible. It is a temporally controlled “on” switch. This distinction should be made clear in the text.

We have updated the text in relevant sections to clarify the inducible switch (but not reversible) nature of our system.

→ Minor updates in several sections across the manuscript

- Lines 54-57: The phrasing here describing the Gutierrez Mena, et al paper is somewhat misleading and in our minds potentially not emphasizing the strengths of the current manuscript. The Gutierrez Mena approach (light-controlled T7 polymerase to induce an antibiotic resistance gene) is almost certainly generalizable (to some extent) although this wasn't tested. So what is novel in the current manuscript is both that generalizability was tested (and indeed four important different resistance mechanisms compared and contrasted) and the Sheets, et al system is irreversible. For some applications this is a disadvantage, but for many others (e.g. one-time induction before production cultures become too saturated, spatial applications, light-sensitive strains) it is an advantage. The language in these sentences could be slightly tweaked to appropriately convey these nuances.

We have modified the text to better represent the Gutierrez Mena work and highlight these differences. We have also described tradeoffs of these designs.

→ See Introduction

→ See Discussion

- Line 66: “minimal basal expression of activated genes” is almost an oxymoron (i.e. if they are activated, it is hopefully not minimally). Would “minimal basal expression in uninduced cells” be more appropriate?

Fixed.

- Line 329: “digital micromirror device” is not defined and there is no reference. It might be that at this point this technology is familiar enough to most groups, but maybe not. It is also true that

there are other technologies that could potentially be used for spatial illumination and that might be worth mentioning.

Fixed – Added a reference to clarify.

- All Figures: At least on our research group's and department's printers, the light gray line tends to disappear when the manuscript is printed. It might be worth switching to a darker color.

Fixed.

- Could the authors comment on why light-induced resistance is never able to reach the same levels of resistance as the positive control strains?

The positive control strains are the native genes, including the native promoter and RBS sequences. Given the different genetic conditions for the Cre system (different promoters, RBSs, and inclusion of a lox site after recombination), a difference in maximum resistance levels is expected. We have clarified this in the text.

→ See Results / Light induction of beta-lactamase resistance

→ See Methods / Strains and plasmids

- The layout of some figures (i.e. Fig. 2 and Fig. 3) is confusing in terms of the organization of B, C, and D. Would it be more clear to separate these elements into columns or rows? Should the legend in the graphs in C also be repeated in the graphs in D?

Figure labels have been updated for clarity.

- One potential downside of this system is that it is not reversible, but it would be possible to use recombinases to develop a reversible switch (with more circuitry). The potential benefits and limitations of this and in general the potential for reversibility should be added to the discussion.

This has been added to the Discussion.

- Line 321: What might require those conditions?

As an example, other plasmids or strains may have existing resistance markers (e.g. one plasmid already uses kanamycin resistance so this needs to be avoided). We have added some language alluding to this need for compatibility with other elements of the design.

- Line 351: Should be “synthetic control of resistance acquisition” rather than gene transfer, as gene transfer wasn't implemented in this paper (although control over HGT would be a very interesting way to explore antibiotic resistance mechanisms and downstream population effects).

Fixed.

- Line 357-359: Do you mean to say here that the enzyme is efficient, but taxing to express at high levels? Or the reaction catalyzed by the enzyme?

From Hernández Lozada et al. (2018), the reaction catalyzed by the enzyme is taxing, although we would expect the high level of protein expression to have an additional burden. We have clarified this in the text.

- Line 424-426: If these experiments were done in an LPA presumably they were done in 24-well plates. What is the volume of cell culture?

We used 1 mL cell cultures and have added this to the text.

- Line 458-459: Should the intensity of the light at the cells also be included here?

Yes, we measured the intensity to be $330 \mu\text{W}/\text{cm}^2$ and have added this to the Methods.

Reviewer #2:

In the manuscript, the authors coupled antibiotic-resistance genes with optogenetics to engineer bacteria with light-induced antibiotic-selective features. They utilized the well-established optogenetic regulator loxP/split-Cre system which can sense blue light and irreversibly activate gene expression. Four commonly used antibiotics with different killing mechanisms and the corresponding resistance genes were engineered to be light-active. They showed that antibiotic-resistance genes could be successfully induced and thus facilitates survival against the antibiotics. Moreover, by manipulating the plasmid backbones, promoters, and RBS, the dynamic range of the antibiotics concentrations the bacteria tolerate can be tuned to the desired value. Then they screened the single-cell growth process in time-lapse with or without light induction, aiming to showcase the possibility of single-cell level behavior of antibiotic resistance acquisition after horizontal gene transfer. At last, they further coupled the system in a fermentation strain with the chemical production pathway so that the cheaters could be killed to improve the yield. The results are overall solid and I have no concerns regarding its technical soundness. However, the demonstration of its potential applications could be further developed. Following are my concerns on this paper:

We thank the reviewer for their positive comments about the technical quality of the work and for their suggestions for improving the applications section of the manuscript.

1. In the fermentation application part, the author was trying to demonstrate that their system can improve the yield. However, the authors did not test the original fermentation strain as the control. The improvements of their engineered system therefore are not characterized. In addition, the fermentation process is a relatively long-term process. Using light induction to kill cheaters could face many issues in real world scenarios, i.e. when to induce, how frequent, and impacts on fermentation strain, etc. To claim its application potentials, more validations are needed.

We agree that robust controls are important for evaluating the work. We note that Reviewer #1 raised a similar concern, and we address these points by now including a constitutively expressed

version of *CpFatB1* as a baseline control (Fig. 6d-e). We found that for OptoCre-*cat*, our inducible system had a greatly improved yield over constitutive expression, and for OptoCre-*tetA*, our inducible system with antibiotic had a similar yield to the constitutive version. However, both of the constitutive versions showed poor growth profiles (Fig. S6). This critical issue indicates that the constitutive versions experience a greater burden than the inducible versions, which will likely lead to escape mutants and reduce the stability of the production system.

→ See Results / Improving octanoic acid production using antibiotic resistance selection section

→ See Fig. 6d-e

→ See Fig. S6

The point about long-term fermentations and timing is also very important. Here, the irreversible nature of the OptoCreVvd system is a potential advantage. Light exposure is only required once for permanent expression of the resistance and production enzymes, simplifying some of the induction timing and impacts. We have updated the text to clarify and highlight this.

→ See Results / Improving octanoic acid production using antibiotic resistance selection section

2. The authors propose in the paper that their system to be used as a model to study the spread of antibiotic resistance genes obtained by horizontal gene transfer, based on the assumption that the light-induced resistance gene expression could be regarded as similar to the acquisition of the resistance genes by horizontal gene transfer. But resistance spread is largely through selection or more subsequent horizontal gene transfer, which is different from transient gene expression increase in the engineered strain after light induction. I would recommend the authors to elaborate how the transient gene expression increase could be an important aspect in studying antibiotic resistance spread. This system has big potentials in investigating one important aspect that could be overlooked. But in current form, this connection is not probed with enough depth.

We thank the reviewer for their note about the potential of our system. As the OptoCreVvd system is irreversible, activated strains would act similarly to strains that have newly acquired a resistance gene, because once the resistance gene has been activated using light it stays on even in the dark, with the permanent genetic switch of the transcription terminator removal. We have added discussion on these points in the text.

→ See Introduction

→ See Discussion

3. The in-text citation numbers should be inside sentences before the punctuations.

Fixed.

4. I suggest adding a few subtitles to the results section to improve the reading experience.

We have added section subtitles.

Reviewers' Comments:

Reviewer #1:

Remarks to the Author:

The authors have addressed most of my major concerns through new experiments and new text.

Specifically:

- The authors did explore increasing terminator strength as a way to improve system performance. They selected a strong synthetic terminator, L3SP21, from the library in Chen, et al 2013. A stronger terminator did not improve leakiness of the system, as shown in Figure S3, therefore the authors did not explore this avenue further. I appreciate the effort on this front, and this is sufficient to demonstrate that increasing terminator strength might not be a route towards tuning the system's basal expression.
- The authors did a convincing job of using sequencing to point to terminator read through as the source of basal resistance in the (- Control) strains, that is, strains without a version of the Cre protein (but with the antibiotic resistance expression construct). More important they have added appropriate discussion of the source of basal resistance in the text.
- I appreciate the authors broadening the context of the introduction to include microbes beyond *E. coli*. A minor point: the focus in the Moreno Morales article was indeed opto-invertase production, however, buried in the supplement (Figure S2) they show that they can use the same ZCRY2 system to drive drug resistance in yeast (nourseothricin), which is to my (admittedly spotty) knowledge still the only example of opto-drug resistance in the widely used *S. cerevisiae* organism. The opto drug resistance was the part of Moreno Morales that I found most relevant to the current study.
- The authors do not characterize the actual level of expression from their various constructs, which I consider a minor deficit. However, they have added new text in the Discussion talking about expression levels and the potential for optimizing these levels in future designs. This is acceptable.
- Line 357: "This growth deficit in the constitutive constructs is problematic, as it may lead to escape mutants and reduce the stability of production strains. Thus, coupling octanoic acid production with resistance selection leads to higher yields of octanoic acid than light induction alone, without the taxing growth deficit associated with continuous production." These, in my opinion, are some of the most interesting observations in the paper and improve the manuscript greatly.
- Figure S2 (showing the ability to tune resistance with light intensity) and Figure S5 (spatial control using the DMD) and the associated discussions are welcome additions to the manuscript. A few comments remain:
 - While the system "could be used with a constitutive OptoCreVvd2" this has never been tested. I still feel that it is worth discussing this point in the Discussion section of the article and making it crystal clear that in the current system (and indeed, in the only system for which there is evidence) there is always a necessary chemical induction step. While the system might work fine with a constitutive OptoCreVvd2, this has not been demonstrated. And indeed, this might require careful tuning of OptoCreVvd2 expression to maintain appropriate background expression levels.
 - Line 138: "Negative control (- Control) cells with only the reporter and no Cre recombinase had nearly identical survival to cells with the full construct grown in the dark, indicating low expression of bla in the uninduced state." This is an overstatement. At least 200 ug looks quite different between (- Control) and Dark. At least in the variance.
 - Line 306: "found 70% recovery for OptoCre-knt and 42% recovery for OptoCre-cat with light exposure (compared to 13% and 4% for cells in the dark, though notably cells in the dark rarely experienced more than one division event, Movie S1-2)." These aren't particularly high recovery rates. Would be interesting to ask what they are if you first expose to light, and then to antibiotic (which I suppose would take some kind of microfluidics setup). This is fine and no additional experiments are needed, but the recovery rates are lower than I would expect.
 - Line 398: "Combining light inducible resistance with a DMD for precise targeting of light would allow for antibiotic selection of a cell line of interest from device outflow, without any chip modification or additional hardware requirements beyond light exposure". What is the utility of this technique relative to photoactivation or photoswitching techniques? (See for example Strack's perspective in Nature Methods "A light switch for targeted genomics")

Reviewer #2:

Remarks to the Author:

The authors addressed all my concerns and it looks great now. Good job making these revisions.

Response to Reviewers
Manuscript # NCOMMS-22-23220B

We appreciate the reviewers' favorable response to our revisions. Below, we summarize the minor changes we have made to the document.

Reviewer #1:

The authors have addressed most of my major concerns through new experiments and new text. Specifically:

- The authors did explore increasing terminator strength as a way to improve system performance. They selected a strong synthetic terminator, L3SP21, from the library in Chen, et al 2013. A stronger terminator did not improve leakiness of the system, as shown in Figure S3, therefore the authors did not explore this avenue further. I appreciate the effort on this front, and this is sufficient to demonstrate that increasing terminator strength might not be a route towards tuning the system's basal expression.
- The authors did a convincing job of using sequencing to point to terminator read through as the source of basal resistance in the (- Control) strains, that is, strains without a version of the Cre protein (but with the antibiotic resistance expression construct). More important they have added appropriate discussion of the source of basal resistance in the text.
- I appreciate the authors broadening the context of the introduction to include microbes beyond E. coli. A minor point: the focus in the Moreno Morales article was indeed opto-invertase production, however, buried in the supplement (Figure S2) they show that they can use the same ZCRY2 system to drive drug resistance in yeast (nourseothricin), which is to my (admittedly spotty) knowledge still the only example of opto-drug resistance in the widely used S. cerevisiae organism. The opto drug resistance was the part of Moreno Morales that I found most relevant to the current study.

We thank the reviewer for highlighting the resistance control in the Moreno Morales work and have updated this in the text.

- The authors do not characterize the actual level of expression from their various constructs, which I consider a minor deficit. However, they have added new text in the Discussion talking about expression levels and the potential for optimizing these levels in future designs. This is acceptable.
- Line 357: “This growth deficit in the constitutive constructs is problematic, as it may lead to escape mutants and reduce the stability of production strains. Thus, coupling octanoic acid production with resistance selection leads to higher yields of octanoic acid than light induction alone, without the taxing growth deficit associated with continuous production.” These, in my opinion, are some of the most interesting observations in the paper and improve the manuscript greatly.
- Figure S2 (showing the ability to tune resistance with light intensity) and Figure S5 (spatial control using the DMD) and the associated discussions are welcome additions to the manuscript.

We appreciate the reviewer's comments and favorable assessment of the changes we made.

A few comments remain:

- While the system “could be used with a constitutive OptoCreVvd2” this has never been tested. I still feel that it is worth discussing this point in the Discussion section of the article and making it crystal clear that in the current system (and indeed, in the only system for which there is evidence) there is always a necessary chemical induction step. While the system might work fine with a constitutive OptoCreVvd2, this has not been demonstrated. And indeed, this might require careful tuning of OptoCreVvd2 expression to maintain appropriate background expression levels.

We have added a reference in the Discussion to the fact that the system requires IPTG induction and that for certain applications avoiding this would be preferable.

- Line 138: "Negative control (- Control) cells with only the reporter and no Cre recombinase had nearly identical survival to cells with the full construct grown in the dark, indicating low expression of bla in the uninduced state." This is an overstatement. At least 200 ug looks quite different between (- Control) and Dark. At least in the variance.

We have updated the text to better represent the data.

- Line 306: "found 70% recovery for OptoCre-knt and 42% recovery for OptoCre-cat with light exposure (compared to 13% and 4% for cells in the dark, though notably cells in the dark rarely experienced more than one division event, Movie S1-2)." These aren't particularly high recovery rates. Would be interesting to ask what they are if you first expose to light, and then to antibiotic (which I suppose would take some kind of microfluidics setup). This is fine and no additional experiments are needed, but the recovery rates are lower than I would expect.

We agree that this is an interesting question, and have added a sentence pointing to future microfluidic experiments that could be used to determine this.

- Line 398: “Combining light inducible resistance with a DMD for precise targeting of light would allow for antibiotic selection of a cell line of interest from device outflow, without any chip modification or additional hardware requirements beyond light exposure” . What is the utility of this technique relative to photoactivation or photoswitching techniques? (See for example Strack's perspective in Nature Methods "A light switch for targeted genomics")

Many of the existing photoactivation and photoswitching techniques, although powerful, have not been shown to work at the bacterial scale. The limited examples that do exist require droplet-based microfluidics or intensive sorting after activation. Additionally, photoswitchable proteins may be diluted out in *E. coli* by the time cells can be selected, making a permanent genetic switch advantageous. We have briefly clarified this in text.

Reviewer #2:

The authors addressed all my concerns and it looks great now. Good job making these revisions.

We thank the reviewer for their favorable view of the improvements we made.